# Multifidelity Multiobjective Optimization for Wake Steering Strategies

**Julian Quick**[1,2], **Ryan N. King**[2], **Garrett Barter**[2], **and Peter E. Hamlington**[1]

[1]University of Colorado, Boulder, CO, USA
[2]National Renewable Energy Laboratory, Golden, CO, USA

**Correspondence:** Julian Quick (julian.quick@colorado.edu)

**Abstract.** Wake steering is an emerging wind power plant control strategy where upstream turbines are intentionally yawed out of perpendicular alignment with the incoming wind, thereby "steering" wakes away from downstream turbines. However, trade-offs between the gains in power production and fatigue loads induced by this control strategy are the subject of continuing investigation. In this study, we present a multifidelity multiobjective optimization approach for exploring the Pareto front of trade-offs between power and loading during wake steering. A large eddy simulation is used as the high-fidelity model, where an actuator line representation is used to model wind turbine blades and a rainflow counting algorithm is used to compute damage equivalent loads. A coarser simulation with a simpler loads model is employed as a supplementary low-fidelity model. Multifidelity Bayesian optimization is performed to iteratively learn both a surrogate of the low-fidelity model and an additive discrepancy function, which maps the low-fidelity model to the high-fidelity model. Each optimization uses the expected hypervolume improvement acquisition function, weighted by the total cost of a proposed model evaluation in the multifidelity case. The multifidelity approach is able to capture the logit function shape of the Pareto frontier at a computational cost only 30% that of the single fidelity approach. Additionally, we provide physical insights into the vortical structures in the wake that contribute to the Pareto front shape.

## 1 Introduction

As wind energy systems have matured, plant-level control has emerged as a new paradigm, where groups of turbines are controlled in coordination to maximize collective power production. This is in contrast to more traditional control strategies, where individual turbines are controlled to maximize their own power production. A potentially promising form of such plant-level control is "wake steering," where upstream wind turbine yaw positions are intentionally misaligned from the incoming wind, "steering" the wake away from downstream turbines. A counter-rotating pair of vortices is generated by the lateral thrust of the wind turbine rotor, which is determined by the yaw offset direction (Fleming et al., 2018; Martínez-Tossas and Branlard, 2020). This allows the performance of wind power plants to be improved by diverting wakes away from downstream turbines

It is speculated that wake steering may produce more power while inducing less total fatigue on all turbines when compared to the baseline strategy of aligning each turbine

with the incoming wind (Howland et al., 2019; Hulsman et al., 2020). However, very few studies have quantified the trade-offs between power and damage. Hulsman et al. (2020) used an actuator line model to train polynomial chaos surrogates for optimization of a weighted sum of power and damage equivalent loads. Yin et al. (2020) present a multiobjective genetic algorithm for the maximization of power and minimization of total thrust. Damiani et al. (2018) performed a detailed analysis of a single wind turbine, finding that negative yaw offsets tended to increase fatigue loading more than positive yaw offsets, but cautioned that this result could not be generalized given the dependence on incident conditions. Other studies have provided additional evidence supporting the notion that positive yaw misalignment results in less edgewise loading of downstream turbine blades than the corresponding negative yaw misalignment strategy (Zalkind and Pao, 2016; López et al., 2020). Wang et al. (2020) demonstrated the potential of individual pitch control to alleviate loads induced from intentionally offsetting the turbine yaw. Van Dijk et al. (2017) used the FLORIS and CCBlade

tools to examine trade-offs between power produced and the edgewise and flapwise fatigue loading induced through wake steering. Lin and Porté-Agel (2020) utilize a large eddy simulation (LES) framework to construct the Pareto set between power and flapwise bending moment loading through a comprehensive parameter sweep. While these studies all provide insights into the trade-offs between power and loading, there is still a need for an efficient optimization algorithm to quantify these trade-offs using computationally-intensive simulations.

Despite its promise, plant level control via wake steering involves complex physics and is challenging to model. Engineering wake models have dubious accuracy when predicting fatigue loading, which higher-fidelity models predict more accurately Rinker et al. (2021). In this study, we propose a multifidelity multiobjective optimization framework to address this challenge and explore trade-offs between power and loading in wake steering strategies. In practice, power and loading will likely be optimized in real time using a singular weighted objective. The relative weights may be decided upon by exploring trade-offs between power and loading using multiobjective optimization to estimate the Pareto frontier. When searching for the Pareto set, an efficient algorithm must balance exploration and exploitation. Several models have been developed that may be used to study the effects of control strategies with various levels of mathematical detail and real-world accuracy (i.e., fidelity) (Annoni et al., 2018; Martínez-Tossas et al., 2019; Hulsman et al., 2020).

Multifidelity optimization exploits the correlation between low- and high-fidelity models to reduce the overall computational cost of optimization. For instance, Andersson and Imsland (2020) present a real-time modifier adaptation approach for wake steering design, where a Gaussian Process (GP) is used to iteratively learn the difference between observed operational data and the predictions of an engineering wake model. Ariyarit and Kanazaki (2017) present a two-objective bifidelity approach that iteratively builds a GP discrepancy function. Huang et al. (2006) and Rajnarayan et al. (2008) employ an augmented expected improvement formulation, including three factors to account for the correlation between the low- and high-fidelity models, the observed error, and the cost ration between the low- and high-fidelity models. It is not always clear when a proposed low-fidelity model is appropriate for use in multifidelity optimization, though it is common to assess candidate low-fidelity models by measuring their correlation with the high-fidelity model (Giselle Fernández-Godino et al., 2019).

The novelty of the present study is the application of this multifidelity technique to wind energy systems, resulting in new insights into wake steering flow physics. The present approach uses the low-fidelity model to first explore the full parameter space, then iteratively builds the low- and high-fidelity model surrogates to gain the most improvement in the Pareto front per model evaluation costs. While this framework is similar to that presented by Ariyarit and Kanazaki

(2017) and Andersson and Imsland (2020), the exact framework outlined here is new and this is the first demonstration of any such approach in the context of wind energy systems.

## 2  Optimization Framework

A Bayesian framework for multiobjective multifidelity optimization is presented. Throughout this section, we assume that minimization of functions is the objective of the optimization procedure (as opposed to maximization).

This study employs GP models to approximate power and loading dynamics. A GP is a collection of random variables, any finite number of which have a joint Gaussian distribution (Rasmussen and Williams, 2006). The simulated power and loads, $f_i$, are approximated using individual Gaussian process surrogate models, $\boldsymbol{g}$, which are defined as

$$g_i(\boldsymbol{\gamma}) \sim \mathcal{GP}\left[\mu_i(\boldsymbol{\gamma}), k_i(\boldsymbol{\gamma}, \boldsymbol{\gamma}')\right],\tag{1}$$

where $\boldsymbol{\gamma}$ is a vector of proposed yaw angles (with dimension equal to the number of turbines), $\boldsymbol{\gamma}'$ is an arbitrary vector of yaw angles, $\mu_i(\boldsymbol{\gamma})$ is the GP mean function, $k_i(\boldsymbol{\gamma}, \boldsymbol{\gamma}')$ is the GP kernel covariance function, and the index $i$ refers to power ($i = 1$) or loading ($i = 2$) objectives.

We perform Bayesian inference on functions by conditioning the GP on a set of observed input-output pairs $\mathcal{D}_i = \{\boldsymbol{\Gamma}, \boldsymbol{Y}_i\}$, where $\boldsymbol{\Gamma} = [\boldsymbol{\gamma}^{(1)}, \boldsymbol{\gamma}^{(2)}, \dots]$ is a matrix of simulated yaw offsets, and $\boldsymbol{Y}_i = [f_i(\boldsymbol{\gamma}^{(1)}), f_i(\boldsymbol{\gamma}^{(2)}), \dots]$ is a vector of observations of simulated power or loading outputs. We use the scikit-learn Gaussian Process implementation (Pedregosa et al., 2011), which is a well-validated open-source project. The power and loading outputs are normalized to have zero mean and unity variance during a preprocessing step. After conditioning on $\mathcal{D}_i$, we obtain Gaussian distributions at test locations $\boldsymbol{\gamma}^*$ with the following posterior estimates of the mean

$$\mu_i(\boldsymbol{\gamma}^* \,|\, \mathcal{D}_i) = \boldsymbol{k}_{i,*}^T \boldsymbol{K}_i^{-1} \boldsymbol{Y}_i\tag{2}$$

and variance

$$\sigma_i^2(\boldsymbol{\gamma}^* \,|\, \mathcal{D}_i) = k_i(\boldsymbol{\gamma}^*, \boldsymbol{\gamma}^*) - \boldsymbol{k}_{i,*}^T \boldsymbol{K}_i^{-1} \boldsymbol{k}_{i,*},\tag{3}$$

where $\boldsymbol{k}_{i,*} = k_i(\boldsymbol{\Gamma}, \boldsymbol{\gamma}^*)$ is a vector and $\boldsymbol{K}_i = k_i(\boldsymbol{\Gamma}, \boldsymbol{\Gamma})$ is a matrix.

The kernel covariance function encodes prior knowledge about structural properties of the underlying signal, such as smoothness, periodicity, and stationarity. In this study, we employ an anisotropic radial basis function kernel (Rasmussen and Williams, 2006) for $k_i$, given by

$$k_i(\boldsymbol{\gamma}, \boldsymbol{\gamma}') = \exp\left[\frac{1}{2}\left(\sum_{j=1}^{\dim(\boldsymbol{\gamma})} \frac{(\gamma_j - \gamma_j')^2}{l_{ij}^2}\right)^{1/2}\right],\tag{4}$$

where the correlation scale, $l_{ij}$, is estimated by maximizing the log-marginal likelihood function (Pedregosa et al., 2011).

## 2.1 Single-Fidelity Approach

In Bayesian optimization, an acquisition function is defined to maximize a metric representing both exploration and exploitation (Shahriari et al., 2015). A popular acquisition function for single objective optimization is the expected improvement (EI), which is the expected value of an improvement function (Zhan and Xing, 2020) with respect to the predicted uncertainty of a GP. This acquisition function is employed by the Efficient Global Optimization (EGO) algorithm (Jones et al., 1998). The improvement function, $I$, quantifies the improvement in the objective function for a new evaluation, as compared to the best sampled objective, and is zero if the new objective does not outperform all of the previously sampled points. This results in

$$I(\boldsymbol{\gamma}) = \max\left[f^* - f(\boldsymbol{\gamma}), 0\right], \tag{5}$$

where $f^*$ is the minimum sampled value and $f(\boldsymbol{\gamma})$ is the function value, which is generally unknown and must be predicted by a GP.

There is a range of potential outcomes from sampling a new point, and the GP framework conveniently estimates this uncertainty. These uncertainties are used to compute the expected value of the improvement function. It is important that the improvement function contain the maximum function; otherwise, there would be no exploration of regions of larger uncertainty. Other acquisition functions available include the knowledge gradient (Ghoreishi and Allaire, 2018), expected quantile improvement (He et al., 2017; Picheny et al., 2013), improved expected improvement (Qin et al., 2017), entropy search (Hennig and Schuler, 2012), and minimization of the predictor (Andersson and Imsland, 2020).

The expected improvement may be extended to a multiobjective context. This is done by introducing a hypervolume function, $H$, which measures the volume of a given Pareto front, $A$, using a reference point, $\boldsymbol{r}$. The expected hypervolume improvement (EHVI), introduced by Emmerich et al. (2006), is the multiobjective counterpart to the expected improvement acquisition function used in the EGO algorithm and is given by

$$EHVI(\boldsymbol{\gamma}, \boldsymbol{g}) = \mathbb{E}_{\boldsymbol{g}(\boldsymbol{\gamma})}\{HVI[\boldsymbol{f}(\boldsymbol{\gamma})]\}. \tag{6}$$

Here, $\mathbb{E}_{\boldsymbol{g}(\boldsymbol{\gamma})}[\cdot]$ represents the expectation with respect to a normal distribution, $\boldsymbol{g}(\boldsymbol{\gamma})$, and is expressed as

$$\mathbb{E}_{\boldsymbol{g}(\boldsymbol{\gamma})}\{HVI[\boldsymbol{f}(\boldsymbol{\gamma})]\} = \int_{-\infty}^{\infty} \int_{-\infty}^{\infty} HVI([P, L])$$
$$\frac{1}{\sigma_1(\boldsymbol{\gamma})\sqrt{2\pi}} e^{-\frac{1}{2}\left[\frac{P-\mu_1(\boldsymbol{\gamma})}{\sigma_1(\boldsymbol{\gamma})}\right]^2} \frac{1}{\sigma_2(\boldsymbol{\gamma})\sqrt{2\pi}} e^{-\frac{1}{2}\left[\frac{L-\mu_2(\boldsymbol{\gamma})}{\sigma_2(\boldsymbol{\gamma})}\right]^2} dP dL, \tag{7}$$

where $\mu_1$ and $\sigma_1$ are the mean and standard deviation, respectively, of the powers modeled by $g_1$, and $\mu_2$ and $\sigma_2$ are

the mean and standard deviation, respectively, of loads modeled by $g_2$. The hypervolume improvement indicator (HVI) function is the multiobjective counterpart to the improvement indicator function. It is given as

$$HVI([P, L]) = H(A \cup \{[P, L]\}) - H(A), \tag{8}$$

where $A$ is an estimated Pareto frontier, $A \cup \{[P, L]\}$ is a new Pareto set that potentially includes $[P, L]$, $H$ is a hypervolume function, which measures the volume spanned by the Pareto set of objective functions relative to a reference point, $\boldsymbol{r}$, which must not be dominated by $A \cup \{[P, L]\}$. The Pareto frontier is defined as the set of all function values that are not strictly dominated by other function values. The formal definition is

$$A = \left\{ \boldsymbol{y}' \in \{\boldsymbol{y} \in \mathbb{R}^{\dim(\boldsymbol{f})} : \boldsymbol{y} = \boldsymbol{f}(\boldsymbol{\gamma}), \boldsymbol{\gamma} \in \Omega_{\boldsymbol{\gamma}}\} : \right.$$
$$\left. \{\boldsymbol{y}'' \prec \boldsymbol{y}', \boldsymbol{y}'' \neq \boldsymbol{y}'\} = \emptyset \right\}, \tag{9}$$

where $\Omega_{\boldsymbol{\gamma}}$ is the set of allowable yaw offsets, $\prec$ denotes Pareto dominance (that is, if $\boldsymbol{y}'' \prec \boldsymbol{y}'$, then $y_i'' \leq y_i'$ for all values of $i$ and $y_i'' < y_i'$ for at least one value of $i$ (Voorneveld, 2003)), and $\emptyset$ is the empty set.

The hypervolume, $H$, measures the extent of the Pareto set as the volume of the Pareto-dominated space bounded by a reference point, $\boldsymbol{r}$, namely,

$$H(A) = \text{Vol}\left(\{\boldsymbol{y} \in \mathbb{R}^{\dim(\boldsymbol{f})} | \boldsymbol{y}' \in A \prec \boldsymbol{y} \text{ and } \boldsymbol{y} \prec \boldsymbol{r}\}\right). \tag{10}$$

In practice, the Pareto set is computed by filtering a set of discrete inputs so that only non-dominated points remain,

$$A = \left\{ \boldsymbol{y}' \in \boldsymbol{Y} : \{\boldsymbol{y}'' \prec \boldsymbol{y}', \boldsymbol{y}'' \neq \boldsymbol{y}'\} = \emptyset \right\}, \tag{11}$$

where $\boldsymbol{Y}$ is a matrix of observed function values. This filters out observed samples that are Pareto dominated by other observed samples.

Although these ideas may also be extended to more objectives, assuming two objectives simplifies the problem. In this case, the observed Pareto set is defined as $A \approx (\boldsymbol{y}^1, \boldsymbol{y}^2, \dots, \boldsymbol{y}^n)$ such that $y_1^1 < y_1^2 < \cdots < y_1^n$. The hypervolume is estimated using rectangular quadrature as

$$H(A) \approx \sum_{i=1}^{n-1} (y_1^{i+1} - y_1^i)(r_2 - y_2^i) + (r_1 - y_1^n)(r_2 - y_2^n), \tag{12}$$

where $n$ is the number of points in the given Pareto set and $r_1$ and $r_2$ are the components of the reference point. In this study, the EHVI is approximated through Monte Carlo simulation by

$$EHVI(\boldsymbol{\gamma}, \boldsymbol{g}) \approx$$
$$\frac{1}{N_s} \sum_{k=1}^{N_s} \left[ H\left(A \cup \{\mathcal{N}[\boldsymbol{\mu}(\boldsymbol{\gamma}), \boldsymbol{\sigma}(\boldsymbol{\gamma})]\}^{(k)}\right) - H(A) \right], \tag{13}$$

where $N_s$ is the number of Monte Carlo samples and $\{\mathcal{N}[\boldsymbol{\mu}(\boldsymbol{\gamma}), \boldsymbol{\sigma}(\boldsymbol{\gamma})]\}^{(k)}$ is draw $k$ from the GP model of power and loading, $\boldsymbol{g}$.

Once the EHVI is estimated, it must be maximized. This is not necessarily trivial, as the EHVI computation is complicated and difficult to vectorize, and the cost of the optimization grows exponentially with the number of design variables. The EHVI optimum may be determined using a grid search, random sampling, direct optimization, or surrogate-based optimization. While a grid search is the most comprehensive option, the latter approaches are more computationally efficient for high dimensional design inputs.

## 2.2 Multifidelity Approach

The multifidelity approach introduces computationally cheaper but lower fidelity representations of the high-fidelity model, which allow for greater control between exploration and exploitation in the Bayesian optimization. Samples of the low-fidelity model are adaptively refined throughout the optimization as a cheap means for exploration of the high-fidelity function space. Throughout this section, we assume a known hierarchy of model fidelities, $(f_k^1, f_k^2, \ldots, f_k^N)$, where $f_k^1$ is the lowest-fidelity model of power/loading, $N$ is the number of different fidelity models, and $f_k^N$ is the highest-fidelity model of power/loading. When a model is evaluated at a point, $\boldsymbol{\gamma}$, we assume that all lower-fidelity models will also be evaluated at this point.

The lowest-fidelity model, $f_k^1$, is approximated using a GP, $g_k^1$, resulting in the following output distribution:

$$g_k^1(\boldsymbol{\gamma}) \sim \mathcal{N}\left[\mu_k^1(\boldsymbol{\gamma}), \sigma_k^1(\boldsymbol{\gamma})\right], \tag{14}$$

where $\mu_k^1$ and $\sigma_k^1$ are the mean and standard deviations, respectively, associated with the lowest-fidelity power/loading model. Higher-fidelity models, $f_k^i(\boldsymbol{\gamma})$, are approximated using additive discrepancy functions that map the next lowest fidelity function, $f_k^{i-1}(\boldsymbol{\gamma})$, to $f_k^i(\boldsymbol{\gamma})$:

$$f_k^i(\boldsymbol{\gamma}) \approx f_k^{i-1}(\boldsymbol{\gamma}) + \delta_k^i(\boldsymbol{\gamma}) \,\forall i > 1, \tag{15}$$

where $\delta_k^i(\boldsymbol{\gamma})$ is the discrepancy function associated with objective $k$ and fidelity $i$,

$$\delta_k^i(\boldsymbol{\gamma}) \sim \mathcal{N}\left[\mu_k^i(\boldsymbol{\gamma}), \sigma_k^i(\boldsymbol{\gamma})\right] \,\forall i > 1, \tag{16}$$

where $\mu_k^i$ and $\sigma_k^i$ are the mean and standard deviations, respectively, modeled by the discrepancy function GP associated with fidelity $i$.

New GPs are defined to extend the EHVI to a multifidelity context. No matter which fidelity is to be sampled next, the ultimate goal is to minimize the highest fidelity functions, so each GP is constructed to predict the high-fidelity outputs. However, GPs associated with lower-fidelity models should not take into account uncertainties associated with higher-fidelity models, as these uncertainties will not be collapsed

if the lower-fidelity model is sampled. Sampling the highest-fidelity model must take all sources of surrogate uncertainty into account, as a high-fidelity model evaluation will be associated with sampling all lower-fidelity models at the same point. So, new GP models, $\boldsymbol{h}^i$, are constructed to predict the high-fidelity output while encoding different uncertainty information. The GPs associated with each fidelity are defined as

$$h_k^i(\boldsymbol{\gamma}) \sim \mathcal{N}\left\{\sum_{j=1}^{N} \mu_k^j(\boldsymbol{\gamma}), \sqrt{\sum_{j=1}^{i}\left[\sigma_k^j(\boldsymbol{\gamma})\right]^2}\right\}. \tag{17}$$

Putting the above formulations together, it is natural to define a multifidelity multiobjective acquisition function as the ratio of EHVI per evaluation cost:

$$J(\boldsymbol{\gamma}, i) = -\frac{EHVI(\boldsymbol{\gamma}, \boldsymbol{h}^i)}{\sum_{j=1}^{i} C_j}, \tag{18}$$

where $J$ is the optimization acquisition function, which is a function of a set of proposed yaw offsets and model fidelity, to be minimized with respect to yaw offsets, $\boldsymbol{\gamma}$, and model fidelity, $i$, in each optimization iteration, and $C_i$ is the computational cost associated with model $i$.

In this study, we examine the bifidelity case ($N = 2$), where

$$h_k^1(\boldsymbol{\gamma}) \sim \mathcal{N}\left[\mu_k^1(\boldsymbol{\gamma}) + \mu_k^2(\boldsymbol{\gamma}), \sigma_k^1(\boldsymbol{\gamma})\right] \tag{19}$$

and

$$h_k^2(\boldsymbol{\gamma}) \sim \mathcal{N}\left\{\mu_k^1(\boldsymbol{\gamma}) + \mu_k^2(\boldsymbol{\gamma}), \sqrt{[\sigma_k^1(\boldsymbol{\gamma})]^2 + [\sigma_k^2(\boldsymbol{\gamma})]^2}\right\}. \tag{20}$$

The bifidelity workflow is visualized in Figure 1. The EHVIs associated with the low-fidelity ($\boldsymbol{h}^1$) and high-fidelity ($\boldsymbol{h}^2$) GP models are maximized. Then, the EHVI per cost is maximized with respect to fidelity in the comparative step, which corresponds to minimizing $J(\boldsymbol{\gamma}, i)$. If the EHVI per cost associated with evaluating the low-fidelity model is greater than the EHVI per cost associated with evaluating the high-fidelity model, the low-fidelity model is evaluated. Otherwise, the high-fidelity model is evaluated. In the figure, $\boldsymbol{h}^1$ is represented by $GP^{\mathrm{LF}}$, $\boldsymbol{h}^2$ is represented by $GP^{\mathrm{HF}}$, $C_1$ is represented by $C_{\mathrm{LF}}$, and $C_2$ is represented by $C_{\mathrm{HF}}$.

## 3 Numerical Approach

This section outlines the numerical approaches used in this study. Section 3.1 presents the flow modeling framework employed. Section 3.2 introduces the specific power and loading objective functions used in this study. Section 3.3 presents the approach used to maximize the multiobjective acquisition function. Section 3.3.1 outlines the sampling approach and Section 3.3.2 presents a correlation analysis used to determine the low-fidelity loading proxy.

## 3.1 Flow Modeling

We use the WindSE framework (National Renewable Energy Laboratory, 2021) to model flow within the wind power plant. We investigate a two-turbine case, with a single wind direction and speed, where the turbines are spaced 7 rotor diameters apart and the wind direction is such that the front turbine directly wakes the back turbine. The large turbine spacing was chosen to ensure that solutions associated with optimal power production were inside of the boundaries of allowable yaw offsets. When turbines are spaced tightly, it is common for the optimal power to be associated with the largest allowable yaw offset of the front turbine, which is a less challenging optimization case. The inflow boundary is modeled using a logarithmic profile with a hub height wind speed of 7.5 m/s. The top, side, and outflow boundaries are specified as no-stress boundaries and the ground is specified as a no-slip boundary. We consider turbine representations of the IEA 3.4 MW reference turbine (Bortolotti et al., 2019), with hub heights of 120 m and rotor diameters of 130 m. The turbine blades are represented as actuator lines with 15 force nodes. This analysis does not consider the turbine nacelle or tower and the turbine blades are modeled as being rigid. There is no turbine controller, and the rotor speed and blade pitch angles are modeled as constants. The domain is represented with a $2260 \times 2000 \times 520$ m$^3$ mesh, corresponding to a 301 s flow-through time. The mesh is refined near the center of the domain and where the turbines are located. A target Courant–Friedrichs–Lewy condition of 0.98 is specified. All simulations were initiated with the same atmospheric conditions.

The simulations solve the filtered conservation of mass and Navier-Stokes equations given by

$$\nabla \cdot (\rho \tilde{\boldsymbol{u}}) = 0, \tag{21}$$

$$\frac{D\tilde{\boldsymbol{u}}}{Dt} = -\frac{1}{\rho}\nabla \tilde{p} + \left(\frac{\mu}{\rho} + \nu_t\right)\nabla^2\tilde{\boldsymbol{u}} + \boldsymbol{\mathcal{F}}, \tag{22}$$

where $D/Dt = \partial/\partial t + \tilde{\boldsymbol{u}} \cdot \nabla$ is the material derivative, $\tilde{\boldsymbol{u}}$ is the velocity, $t$ is time, $\boldsymbol{x}$ is the spatial location, $\nabla$ is the spatial gradient, $\rho$ is the density, $\tilde{p}$ is the pressure, $\mu$ is the dynamic viscosity, $\nu_t$ is the turbulent viscosity, and $\boldsymbol{\mathcal{F}}$ is the turbine forcing. The density is specified as $\rho = 1$ kg m$^{-3}$ and the dynamic viscosity is specified as $\mu = 1.8 \times 10^{-5}$ kg m$^{-1}$ s$^{-1}$. The turbulent viscosity, $\nu_t$, is modeled using the Smagorinsky–Lilly LES model as

$$\nu_t = C_s^2\Delta^2|\boldsymbol{S}|, \tag{23}$$

where $C_s = 0.17$, $\Delta$ is the grid cell size, and $\boldsymbol{S}$ is the strain rate tensor given by

$$\boldsymbol{S} = \frac{1}{2}\left[\nabla\tilde{\boldsymbol{u}} + (\nabla\tilde{\boldsymbol{u}})^T\right], \tag{24}$$

and $|\boldsymbol{S}| = (2\boldsymbol{S}:\boldsymbol{S})^{1/2}$.

The turbine forcing is computed as

$$\boldsymbol{\mathcal{F}}(\boldsymbol{x}) = \sum_{k=1}^{\dim(\boldsymbol{\gamma})}\sum_{b=1}^{3}\sum_{j=1}^{N_{\text{nodes}}}\boldsymbol{f}_{kbj}(\boldsymbol{x})\frac{1}{\pi^{3/2}\epsilon^3}\exp\left[-\frac{|\boldsymbol{d}_{kbj}(\boldsymbol{x})|^2}{\epsilon^2}\right], \tag{25}$$

where $N_{\text{nodes}}$ is the number of actuator nodes per blade, $\epsilon$ is the characteristic width of the actuator forces (specified as 2 m in this study), $\boldsymbol{d}_{kbj}(\boldsymbol{x})$ is the distance from node $j$ associated with blade $b$ and turbine $k$, and $\boldsymbol{f}_{kbj}(\boldsymbol{x})$ is the actuator force on node $j$ associated with blade $b$ and turbine $k$, which is computed based on the wind speed, angle of attack, and the reference airfoil lift and drag coefficients, as well as a tip loss correction, as described by Allen et al. (2022).

Low- and high-fidelity models were developed for this study using the WindSE framework. A Cartesian discretization of the computation domain is specified, where the grid is refined twice in the wake region as well as near the turbine rotors. Each high-fidelity simulation is run to 1,200 s using Taylor-Hood elements (Ern and Guermond, 2004). The power and loading results only use information from the final 600 s of simulation time. These time parameters were justified by comparing power and loading computed over time intervals of 600-900 s and 900-1,200 s, resulting in relative differences of only 2.6% for power and 4.2% for loading when $\boldsymbol{\gamma} = (15°, 0°)$. These time parameters were chosen to obtain a reasonably efficient optimization problem for the present demonstration, accepting the possibility that the optimization results could be slightly different if different time parameters were chosen. For example, using a time interval

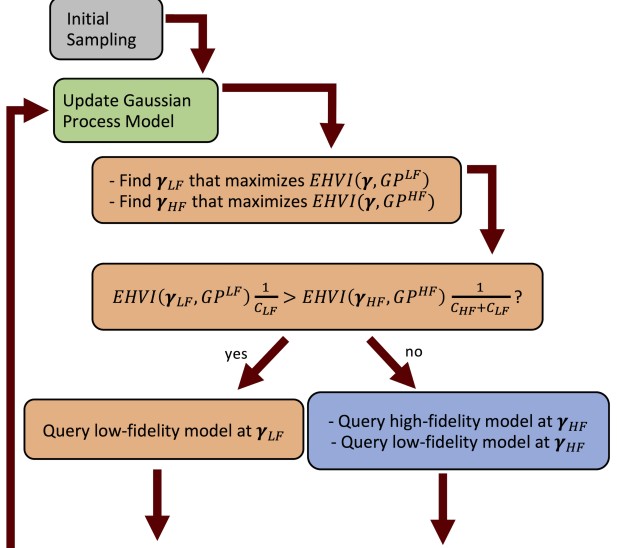

**Figure 1.** Workflow visualization for the bifidelity optimization case. $\boldsymbol{h}^1$ is represented by $GP^{\text{LF}}$, $\boldsymbol{h}^2$ is represented by $GP^{\text{HF}}$, $C_1$ is represented by $C_{\text{LF}}$, and $C_2$ is represented by $C_{\text{HF}}$.

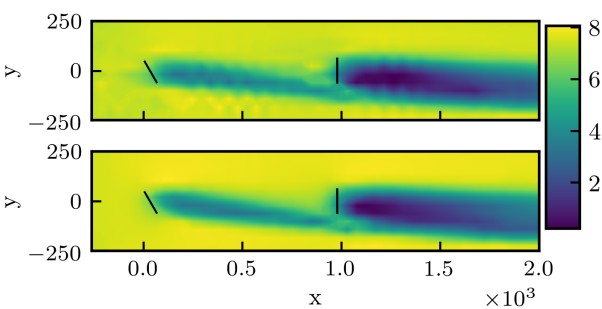

**Figure 2.** Time averaged velocity magnitude fields at the turbine hub height associated with the low- and high-fidelity models. The top panel shows the flow field associated with the low-fidelity model and the bottom panel shows the flow field associated with the high-fidelity model. In both cases, the front turbine is offset by 30° and the back turbine has no yaw offset. Brighter colors correspond to faster velocity magnitudes. Turbine positions are shown with dark lines.

from 1,200-1,800 s, instead of the present 600-1,200 s resulted in changes to the hypervolumes of the Pareto fronts discovered by the single-fidelity and multifidelity optimizations of less than 0.5%. The low-fidelity model was selected using the same grid as the high-fidelity model, but runs to 400 simulation seconds and uses piecewise linear elements. The power is averaged after 300 s and the loading is estimated using the front and back turbine moments past 300 s. This cut-in time and the total low-fidelity model simulation time were selected to avoid initial transient effects while minimizing the computational cost of the low-fidelity simulation. Using 8 processors, the high-fidelity model was measured to take 5.4 hours to run and the low-fidelity model was measured to take 0.24 hours to run. This corresponds to a cost ratio of approximately 0.05. The flow fields produced by the low- and high-fidelity models are compared in Figure 2.

## 3.2   Objective Functions

The objective of the optimization is to minimize negative power, $-P$, and loading, $L$, with respect to each turbine yaw offset such that the yaw offset angles, $\boldsymbol{\gamma}$, are bounded between $-30°$ and $30°$.

Power and loading are quantified using the actuator line model results, discarding an initial transient period. Power is computed as the average total power after the initial transient period. While there are several ways to quantify loading, this study provides a demonstration of the optimization framework by summarizing the time history of the flapwise bending moment of one blade in the front and back turbines after the same initial transient period. Using the high-fidelity model, loading is computed as the sum of damage equivalent loads (DELs) (International Electro-technical Commission, 2015) associated with the front and back turbine flapwise bending moments. The power and flapwise bending moment

are computed from the actuator force as

$$\mathcal{P}^k(t) = \omega \sum_{b=1}^{3} \sum_{j=1}^{N_{\text{nodes}}} r_j \left( \boldsymbol{f}_{kbj} \cdot \hat{\boldsymbol{n}}_{\theta,k} \right) \qquad (26)$$

and

$$M^k(t) = \sum_{j=1}^{N_{\text{nodes}}} r_j \left( \boldsymbol{f}_{k1j} \cdot \hat{\boldsymbol{n}}_{n,k} \right), \qquad (27)$$

where $\mathcal{P}^k(t)$ is the power associated with turbine $k$, $M^k(t)$ is the flapwise bending moment associated with turbine $k$, $\omega$ is the angular speed of the rotor (which is a constant 11.6 rotations per minute in this study), $r_j$ is the radial location associated with node $j$, $\hat{\boldsymbol{n}}_{n,k}$ is the unit vector orientated outward from the rotor plane associated with turbine $k$, and $\hat{\boldsymbol{n}}_{\theta,k}$ is the unit vector oriented in the direction of rotation of turbine $k$.

The average power production of each turbine is computed as

$$\mathcal{P}^{k,\text{avg}} = \frac{1}{t_f - t_0} \int_{t_0}^{t_f} \mathcal{P}^k(t)dt, \qquad (28)$$

where $\mathcal{P}^{k,\text{avg}}$ is the average power associated with turbine $k$, $t_0$ is the initial time considered, and $t_f$ is the final time of the data set. The total power, measured in megawatts, is computed as the sum of the powers produced by each wind turbine,

$$P = \sum_{k=1}^{\dim(\boldsymbol{\gamma})} \mathcal{P}^{k,\text{avg}}. \qquad (29)$$

Each DEL is computed using the rainflow counting algorithm as

$$DEL(M^k) = \left( \sum_{i=1}^{100} R_i^m \frac{c_i}{\Delta t} \right)^{1/m}, \qquad (30)$$

where $i$ loops through each cycle found using the rainflow counting algorithm, $R_i$ is a load range, $c_i$ is the number of cycles associated with the $i$th range bin of the moment load spectrum, $\Delta t$ is the time elapsed in seconds, and $m$ is the Wöhler Exponent. In this study, $m$ is set as 10, and $R_i$ and $c_i$ are computed using the fatpack Python package (Frøseth and Capponi, 2021), utilizing 100 loading bins. The loading objective is computed as the sum of the flapwise bending moment DELs associated with the front and back turbines as

$$\hat{L}^{\text{HF}} = \sum_{k=1}^{\dim(\boldsymbol{\gamma})} DEL(M^k) \qquad (31)$$

The loading objective, $L$, is normalized to be negative and on a similar scale to power, as

$$L = \hat{L}/10^7 - 10 \qquad (32)$$

where $\hat{L}$ is the load prior to normalization, which is computed in Newton-meters. This normalization was chosen to ensure that power and loading are of similar scale and that both are negative. Because the EHVI is an area produced by the two objectives, we do not expect different values in this scaling function to affect the results of maximizing the acquisition function, provided that all sampled objective values are always less than the associated reference value.

## 3.3 Optimization Implementation

Here we use a simple optimization approach for simplicity of demonstration. In each iteration, the $l$ correlation scale parameter in Eq. (4) is selected based on the maximum likelihood function (Pedregosa et al., 2011), with a lower bound of $5°$ and an upper bound of $30°$. The reference point, $r$, in Equation 12 is specified as $(0°, 0°)$. In our formulation, we minimize $J(\gamma, l)$ using a grid-based search for the maximum value. The grid is evenly spaced with 31 inputs per yaw offset dimension. Individual grids are considered for all values of $l$. The $EHVI$ is computed using Monte Carlo sampling with 1,200 samples taken from the GP. The EHVI may also be computed through numerical quadrature (Emmerich et al., 2011; Hupkens et al., 2015). When dealing with inputs of larger dimension, the EHVI may be maximized using an optimization algorithm, such as a genetic algorithm or the EGO approach. After the optimization, the Pareto set was refined using B-spline interpolation in SciPy (Virtanen et al., 2020).

### 3.3.1 Initial Sampling

Initial sampling points are selected using a heuristic approach, where an assumed kernel is used to progressively minimize the standard deviation of the predictor. The simplest approach to initializing the optimization procedure is to randomly sample the low-fidelity and discrepancy functions. Random initial sampling may affect the optimization results, so a deterministic and symmetric sampling strategy is used as a test case. An isotropic kernel is used with a correlation scale of $10°$. The GP model is initialized with the point $(0°, 0°)$. A 100 by 100 grid of inputs ranging between $-30°$ and $30°$ degrees is used to find the next point that minimizes uncertainty in the predicted variance. This process was repeated iteratively to generate 100 points. These points were used to naively estimate the optimal power and loading and Pareto hypervolume as a reference. The first 5 points were used as the initial high-fidelity samples and the first 20 points are used as the initial low-fidelity samples. This heuristic sampling approach is also used to generate 100 samples for use in a correlation analysis and as a naive, baseline approach to searching for the Pareto set.

### 3.3.2 Low-Fidelity Loading Model

We considered several different low fidelity model forms for loading, and selected the one with the highest correlation to the high fidelity model, as that is known to result in the best multi-fidelity performance. We used 100 samples obtained using the heuristic sampling method described in Section 3.3.1 to test the correlation. In practice, this correlation test would not be part of the optimization procedure and reasonably accurate models would be identified based on past experience and/or expert opinion. The correlation analysis revealed a correlation of 0.976 between the low- and high-fidelity power predictions. Using the DEL as the load proxy in the low-fidelity model yielded a low correlation between the low- and high-fidelity models. We explored other potential loading proxies—applying the proxy to both the front and back turbine moment histories then summing the results—and the results of each correlation analysis are presented in Table 1. We used the proxy associated with the highest-measured correlation, namely

$$\hat{L}^{\mathrm{LF}} = \mu_t \left[ M^{\mathrm{front}}(t) \right] + 5\sigma_t \left[ M^{\mathrm{front}}(t) \right] + \\ \mu_t \left[ M^{\mathrm{back}}(t) \right] + 5\sigma_t \left[ M^{\mathrm{back}}(t) \right], \quad (33)$$

where $\mu_t$ and $\sigma_t$ are the mean and standard deviation, respectively, with respect to time. The DEL is purposely replaced with lower-order moment functions to avoid the influence of the spurious oscillations caused by the low-fidelity loading model. The DEL is essentially a high-order moment, which is especially susceptible to these oscillations. The lower-order moments in Eq. (33) were less susceptible to the spurious oscillations, which is why larger correlations ares observed.

## 4 Results and Discussion

### 4.1 Pareto Set Computation

The convergence of the single-fidelity and multifidelity optimization approaches are compared in Figure 3. The dashed lines show the hypervolume, best-sampled load, and best-sampled power found from 100 sampled points using the heuristic sampling approach, which the single-fidelity and multifidelity approaches both outperformed. The EHVI associated with the multifidelity approach was generally lower than the EHVI associated with the single-fidelity approach. The multifidelity approach took less than a third as much total time to estimate the optimal power and loading compared to the single-fidelity approach. The optimal power is achieved with $26°$ offset in the front turbine and $2°$ offset in the back turbine. The optimal loading is achieved with $22°$ offset in the front turbine and $-30°$ offset in the back turbine. We also performed several shorter optimizations as part of the development process using random initial samples, confirming that the multifidelity approach consistently determined the correct hypervolume in fewer iterations than the single fidelity approach.

**Table 1.** Correlations observed between high-fidelity DEL and different loading proxies of the low-fidelity model using 100 heuristic samples.

| Proxy | $DEL$ | $\mu$ | $\sigma$ | $\mu+\sigma$ | $\mu+2\sigma$ | $\mu+3\sigma$ | $\mu+4\sigma$ | $\mu+5\sigma$ | $\mu+6\sigma$ |
|---|---|---|---|---|---|---|---|---|---|
| Correlation | 0.742 | 0.103 | 0.800 | 0.479 | 0.745 | 0.857 | 0.892 | 0.899 | 0.896 |

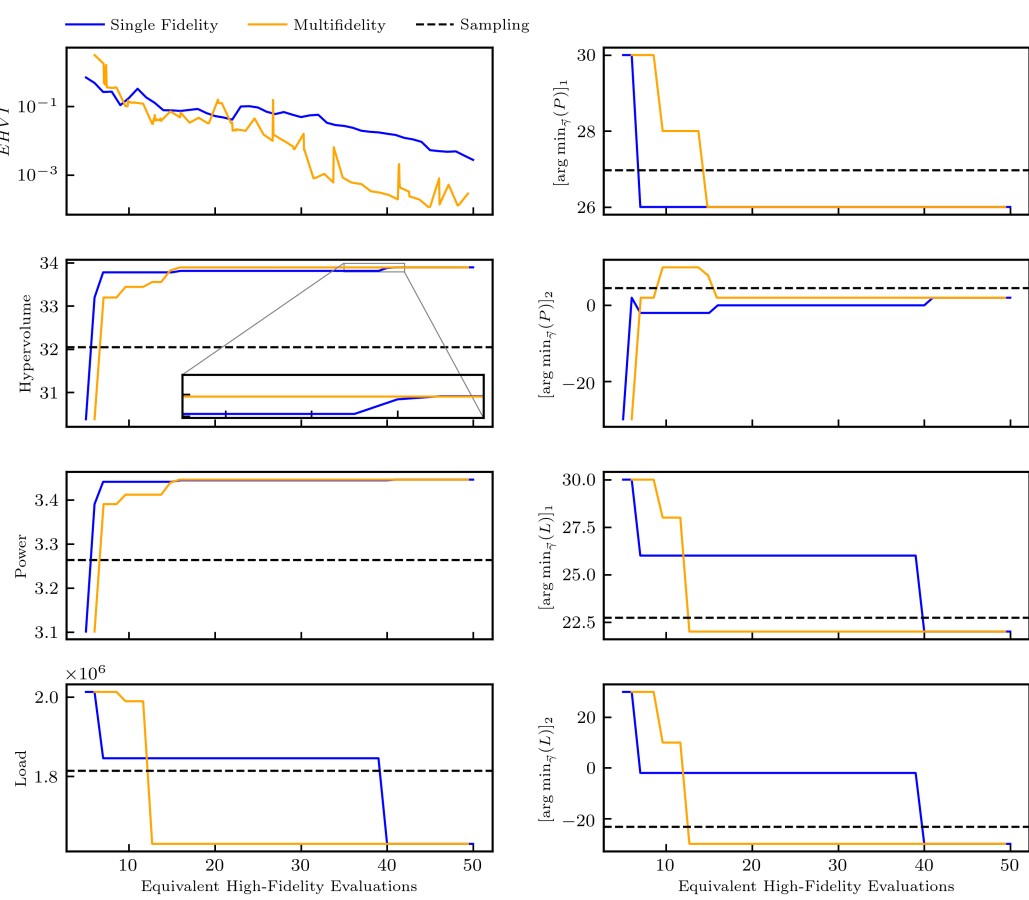

**Figure 3.** Convergence history of the single-fidelity and multifidelity approaches. The left plots show the EHVI, hypervolume, and best-observed power and loading. The right plots show the yaw configurations associated with the best-observed power and loading.

Figure 4 compares the Pareto sets found using 50 equivalent high-fidelity evaluations using the single-fidelity and multifidelity approaches. The multifidelity approach captures five Pareto points and the single-fidelity approach captures four Pareto points. The estimated Pareto sets are very similar, although the single-fidelity approach captures more of the Pareto set close to the optimal power and the multifidelity approach captures more of the Pareto set close to the optimal loading. The results of the single-fidelity and multifidelity approaches are combined to show a single Pareto set, which has a shape similar to a logit function.

Although the primary goal of the present study is to develop and demonstrate a multifidelity multiobjective optimization framework, once several of the points in the Pareto set are identified the set can be further refined using a grid search. As a demonstration of this additional step, the Pareto

set resulting from the combination of the single-fidelity and multifidelity approaches was interpolated to create refinement samples using B-spline interpolation (Virtanen et al., 2020) with 10 interpolation points. To pick up more of the Pareto set, this interpolation was offset in the $\gamma_1$ direction by $-2°$, $-1°$, $1°$, and $2°$ when creating the input refinement set.

The Pareto set resulting from these additional refinement samples is visualized in Figure 5. The resulting Pareto set has three more points than the Pareto set found combining the single-fidelity and multifidelity approaches. The optimization algorithm did not originally fill in these points because they reside in a relatively flat portion of the Pareto set (i.e., $dP/dL$ is small), where adding points would not be expected to increase the Pareto set hypervolume based on the rectangular quadrature employed in this study. Adding these points increased the hypervolume of the discovered Pareto set by a

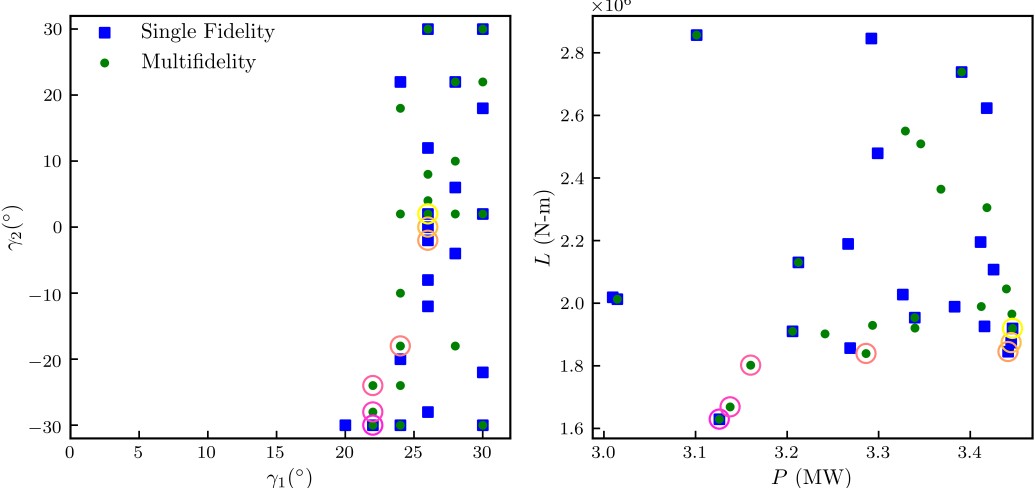

**Figure 4.** Sampled inputs and outputs associated with power greater than 3 MW and loads less than 3 MN-m. Points associated with the single-fidelity approach are shown using blue square markers, and points associated with the multifidelity approach are shown using green circular markers. A Pareto set constructed from the single-fidelity and multifidelity results is highlighted with hollow circles, where darker (magenta) circles correspond to Pareto points with lower loads and lower powers.

very small amount, on the order of 0.002%. Even when using such a refined set of inputs, there are several points where the Pareto set jumps from one yaw position to another; attention would be needed for an operator to control the turbines to operate along the Pareto set.

The multifidelity approach was successful in quantifying the trade-offs between loading and power, and was shown to be more efficient than its single-fidelity counterpart. From the presented results, we find that loading may be reduced by 4% while only reducing the optimal power by 0.3%. The accuracy of the single-fidelity and multifidelity GP models are quantified using a leave-one-out analysis in the Appendix. Table 2 shows the power and front and back turbine DELs associated with several strategies. Slight adjustments to the back turbine angle result in substantial differences in the back turbine loading. These small changes in yaw position adjust the turbine thrust away from the flow, reducing the total thrust imparted on the back turbine.

### 4.2  Flow Physics Insights

Figure 6 shows the flow fields associated with neutral, $-30°$, and $+30°$ yaw offsets in the front turbine, with the back turbine aligned with the wind direction. When the front turbine is offset, two structures are produced: a pair of counter-rotating vortices as well as a coherent structure that is drawn from the boundary layer. The direction of vortex rotation is determined by the direction of thrust the turbine imparts on the incoming air. Induced vortices generally rotate in the opposite direction from the blades that generated them, and the location of the vortices is determined by their rotational direction and the direction of blade rotation. The upper vortex

associated with the positive yaw offset is lower in elevation than the upper vortex produced by the negative yaw offset. The upper vortex also drifts less in the crossflow direction when using the positive yaw offset than when using a negative yaw offset. The bottom vortex drifts similarly in both the positive and negative offset cases. All this amounts to a larger and more extreme velocity deficit encroaching on the back turbine when using the negative yaw offset, rather than the positive yaw offset, resulting in more loading and less power.

Time-averaged flow fields associated with the optimal power, $\boldsymbol{\gamma} = (26°, 2°)$, and optimal loading, $\boldsymbol{\gamma} = (22°, -30°)$, solutions are compared in Figure 7, which shows vertical slices of the flow field before the flow reaches the back turbine. The lesser front turbine yaw offset angle in the $\boldsymbol{\gamma} = (22°, -30°)$ case results in less lateral movement of the wake, and the wake structure has greater overlap with the back turbine than in the $\boldsymbol{\gamma} = (26°, 2°)$ case. The stronger vortical motion resulting from the $\boldsymbol{\gamma} = (26°, 2°)$ case results in the boundary layer being convected further inwards. This boundary layer structure also impacts the back turbine less in the $\boldsymbol{\gamma} = (22°, -30°)$ case because of the reduced back turbine projected area. As the wake convects past the back turbine, additional vorticity is added to the flow. In the $\boldsymbol{\gamma} = (22°, -30°)$ case, the boundary layer structure appears to be pushed back down by the rotation of the bottom vortex.

Figure 8 shows time histories of the flapwise bending moment associated with the front and back turbines for various wake steering strategies. The spikes in the back turbine loading history are caused by the wake impacting the back turbine. When $\boldsymbol{\gamma} = (-30°, 0°)$, the back turbine shows greater downward spikes in loading than when $\boldsymbol{\gamma} = (30°, 0°)$, be-

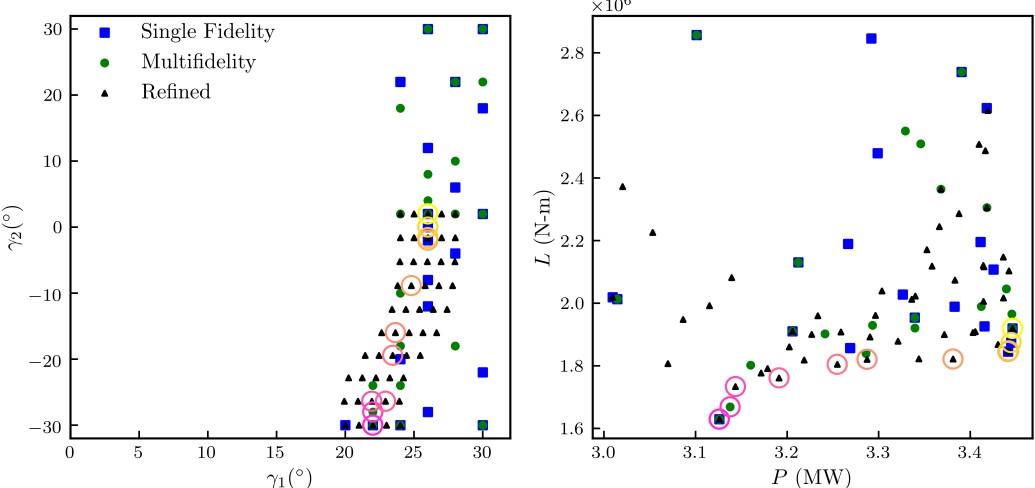

**Figure 5.** Sampled inputs and outputs associated with power greater than 3 MW and loads less than 3 MN-m. Points associated with the single-fidelity approach are shown using blue square markers and points associated with the multifidelity approach are shown using green circular markers. Refinement points are shown as black triangles. A Pareto set constructed from the single-fidelity, multifidelity, and refinement samples is highlighted with hollow circles, where darker (magenta) circles correspond to Pareto points with lower loads and lower powers.

**Table 2.** Observed power and loads for various yaw configurations

| $\gamma_1$ (°) | $\gamma_2$ (°) | Power (MW) | Load (M-Nm) | Front Turbine Power (MW) | Back Turbine Power (MW) | Front Turbine DEL (M-Nm) | Back Turbine DEL (M-Nm) |
|---|---|---|---|---|---|---|---|
| 30  | 0   | 3.24 | 3.53 | 1.84 | 1.40 | 0.78 | 2.75 |
| −30 | 0   | 2.99 | 5.59 | 1.84 | 1.15 | 0.93 | 4.66 |
| 26  | 2   | 3.45 | 1.92 | 2.12 | 1.33 | 0.61 | 1.31 |
| 26  | 0   | 3.44 | 1.87 | 2.12 | 1.32 | 0.61 | 1.27 |
| 26  | −2  | 3.44 | 1.85 | 2.12 | 1.32 | 0.61 | 1.24 |
| 26  | −30 | 3.25 | 2.18 | 2.12 | 1.13 | 0.61 | 1.57 |
| 22  | 0   | 3.26 | 3.55 | 2.21 | 1.05 | 0.52 | 3.03 |
| 22  | −30 | 3.13 | 1.63 | 2.22 | 0.91 | 0.52 | 1.11 |

cause of the greater velocity deficit discussed above. The $\gamma = (26°, 0°)$ case yields smaller downward spikes associated with the back turbine loading than when $\gamma = (30°, 0°)$, because the strength of the counter-rotating vortices is such
that the structure convected from the boundary layer does not impact the back turbine as adversely. The $\gamma = (22°, 0°)$ offset case has larger downward spikes in the back turbine loading than the $\gamma = (26°, 0°)$ offset case, because the latter steers the wake further away from the back turbine. When
the back turbine is offset to $−30°$, the back turbine thrust and associated moments are generally reduced. With this extreme back turbine yaw offset, there is less variation in the back turbine loading when the front turbine is offset by $22°$ than when it is offset by $26°$, because the former case results
in greater variation of velocity across the back turbine rotor plane. These results are specific to the specified spacing between turbines and atmospheric conditions.

## 5  Conclusions

This paper has demonstrated a multifidelity multiobjective optimization approach for wake steering strategies. Actua- 20 tor line simulations were carried out using the WindSE tool, using a coarser simulation as the low-fidelity model. The high-fidelity loading was characterized as the sum of flapwise bending moment DELs on blades on the front and back turbines. Due to oscillations in the low-fidelity simulations, 25 characterizing the low-fidelity loading with a DEL resulted in a relatively low correlation between the low- and highfidelity loading predictions, so a different low-fidelity surrogate was developed with a higher correlation.

The multifidelity multiobjective optimization approach 30 was effective in exploring the trade-offs between loading and power when developing a wake steering design. Convergence was achieved in the multifidelity optimization case after approximately 30% as many equivalent high-fidelity model evaluations as in the single-fidelity case. Future work 35

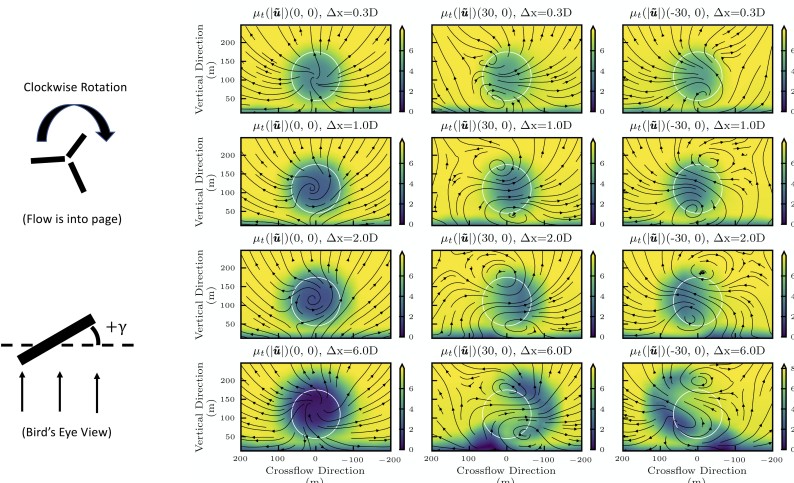

**Figure 6.** Flow fields associated with the extreme and neutral offsets in the front turbine, viewed from upstream. The $\Delta x$ term indicates the distance downstream from the front turbine in terms of rotor diameters. Brighter colors show faster velocities. Streamlines show the direction of the crossflow and vertical velocity components. In each plot, the vertical and crossflow location of the back turbine is shown as a white circle. The turbines rotate clockwise when viewed from upstream. A diagram is shown on the left depicting the direction of positive yaw offset when viewing the turbine from above.

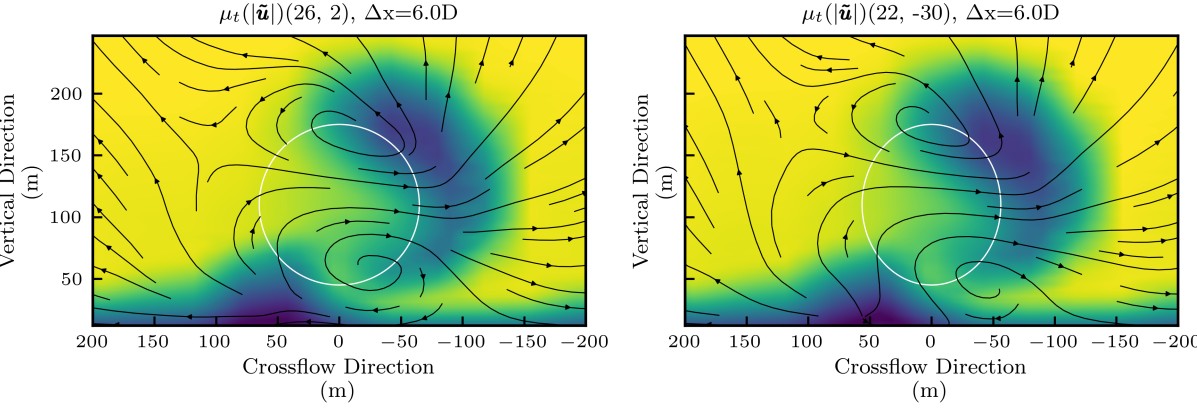

**Figure 7.** Time-averaged flow fields associated with the optimal power (left) and loading (right) found by the optimization, viewed from upstream, 6 rotor diameters (top) away from the front turbine, one rotor diameter upstream of the downstream turbine. Brighter colors indicate faster velocity magnitudes. In each plot, the vertical and crossflow location of the back turbine are shown as a white ellipse.

should apply this approach and a low-fidelity loading function to more complex wind plant layouts to confirm their effectiveness. Exploring the solutions in the final Pareto sets guided insights into the fundamental flow physics. Given the specified turbine spacing and atmospheric conditions, a positive front turbine yaw offset is more effective at reducing loading and increasing power than a negative yaw offset because the counter-rotating vortices associated with the negative front turbine yaw offset produce a greater velocity deficit in the downstream wake. The boundary layer is convected by the counter-rotating vortices, adversely affecting loading, and this may be avoided using less-extreme front turbine yaw offsets. Slightly modifying the back turbine yaw offset

reduced loading by 4% and only reduced power by 0.3%. Greater offsets in the back turbine also led to less overall loading, with significantly less power generation.

It is well known that yaw position errors can adversely affect the performance of wake steering strategies. This is especially true when it comes to turbine loading. A 30° yaw offset is already an aggressive strategy, and unfavorable yaw position errors may result in even more aggressive yaw offsets in practice. Yaw offset errors are generally extreme in lower wind speeds, which is when wake steering strategies are most efficient at increasing power. Previous work has examined the potential of considering yaw error uncertainties in the wake steering optimization problem

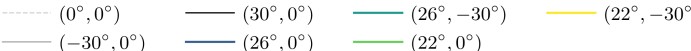

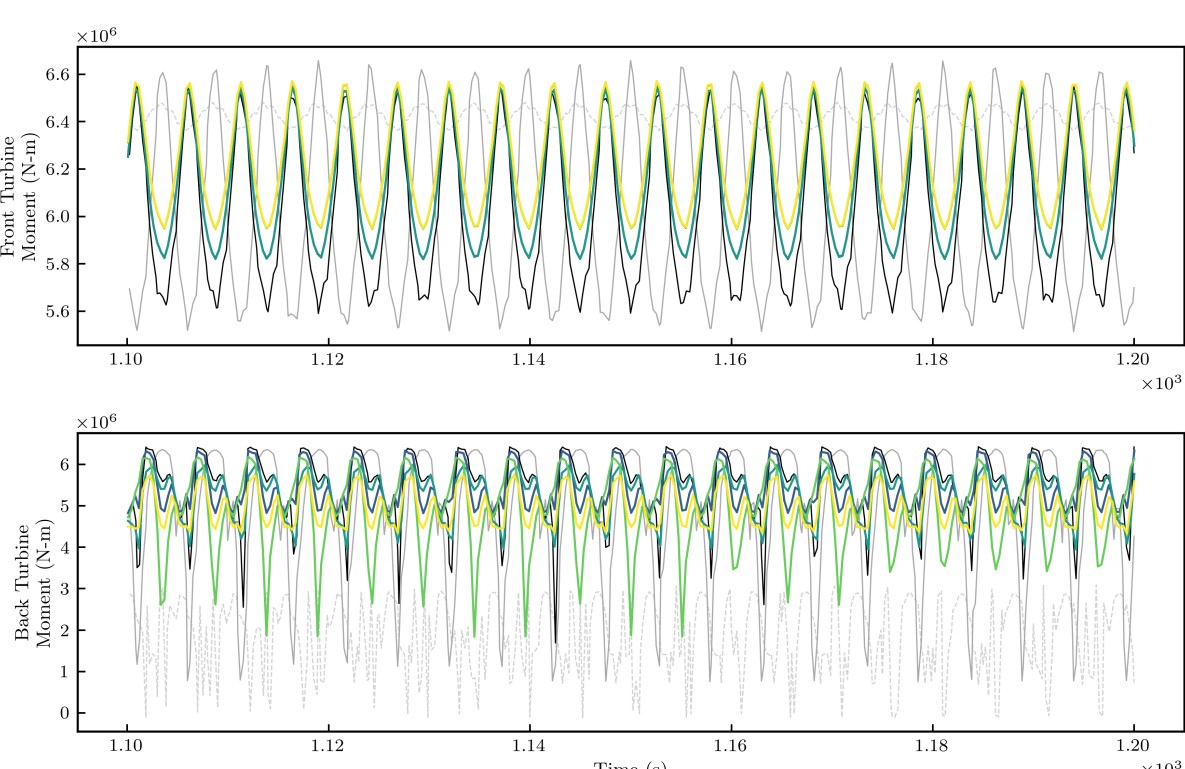

**Figure 8.** Loading histories associated with different yaw offset strategies (values of $\gamma$).

(Quick et al., 2017, 2020). The multifidelity optimization approach presented in this paper could conceivably be extended to optimization under uncertainty, using the final GP models to propagate yaw position uncertainty, and potentially even modifying the EHVI definition to include uncertainty information. Incorporating uncertainty will likely change the shape of the discovered Pareto front.

A drawback of the presented approach is that it requires sequential high-fidelity model evaluations. In practice, it is often feasible to evaluate a high-fidelity model several times in parallel, and the greatest expense is time needed to run the optimization. This framework may be extended to allow for parallel function evaluations. A simple approach is to use predictions of the GP as stand-ins for future model evaluations, iteratively using these points to construct the next iteration of the GP and the associated EHVI (Ginsbourger et al., 2010). Yang et al. (2019) propose dividing the input space into separate regions for parallelization of EHVI optimization. Another intuitive approach could be to include refinement points during each iteration. Refinement points could be selected using the Pareto set predicted by the GP models or interpolated along the observed Pareto set. Care should also be taken when applying this method to ensure conver-

gence of the Pareto set with respect to convergence of the underlying simulation.

In future work, this framework can be applied to a larger array of turbines using more realistic control strategies with different turbine spacings and atmospheric conditions. While considering more turbines presents additional complications in maximizing the EHVI, we anticipate there will be even greater cost savings from the multifidelity approach as the number of turbines increases. Additionally, the framework can be extended to allow for optimization under uncertainty, as it is not realistic to assume perfect control of wind turbine yaw positions. Finally, the framework can incorporate more lower-fidelity models and be combined with layout optimization to realize the full benefits of multifidelity multiobjective wake steering optimization.

## 6   Acknowledgments

This work was authored in part by the National Renewable Energy Laboratory, operated by Alliance for Sustainable Energy, LLC, for the U.S. Department of Energy (DOE) under Contract No. DE-AC36-08GO28308. Funding provided by the U.S. Department of Energy Wind Energy Technologies

Office. The views expressed in the article do not necessarily represent the views of the DOE or the U.S. Government. The U.S. Government retains and the publisher, by accepting the article for publication, acknowledges that the U.S. Government retains a nonexclusive, paid-up, irrevocable, worldwide license to publish or reproduce the published form of this work, or allow others to do so, for U.S. Government purposes. This research was performed using computational resources sponsored by the Department of Energy's Office of Energy Efficiency and Renewable Energy and located at the National Renewable Energy Laboratory.

## 7 Appendix

A leave-one-out analysis was performed to assess the accuracy of the single-fidelity and multifidelity GP models. For each point considered, the model was trained using the remaining points available, excluding the point of interest. Then, the accuracy of the prediction was quantified by comparing it to the observed value. The single-fidelity and multifidelity approaches were analyzed using the data associated with the optimization case study. When assessing the accuracy of the multifidelity model, the low- and high-fidelity samples associated with each point considered in the leave-one-out analysis were removed. Figure 9 shows results of the leave-one-out analysis associated with the single-fidelity GP. Figure 10 shows the results of the leave-one-out analysis associated with the multifidelity GP.

Based on these results, both GPs served as reasonably accurate surrogates. Many of the sampled errors are less than 0.1 MW and 0.1 MN-m, particularly in the region of the discovered Pareto set, which correspond to 3% of the maximum power and 6% of the minimum loading, respectively. The multifidelity approach yielded higher maximum errors and lower minimum errors than the single-fidelity approach. This analysis focused on the final results of the optimization, and we generally expect the leave-one-out errors to shrink as the optimization progresses.

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

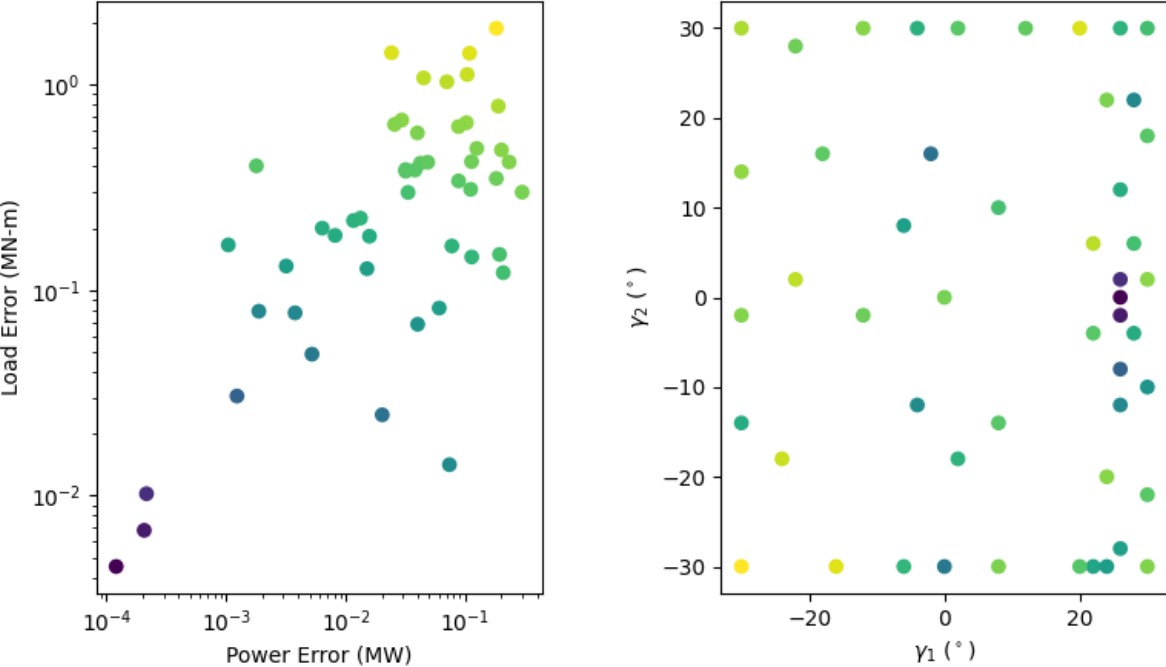

**Figure 9.** Results of the single-fidelity leave-one-out analysis. The left panel shows the leave-one-out prediction errors associated with power and loading, and the points are colored by the sum of both errors. The same points are plotted in the right panel, showing their associated $\gamma_1$ and $\gamma_2$ values.

Jones, D. R., Schonlau, M., and Welch, W. J.: Efficient global op-
    timization of expensive black-box functions, Journal of Global
    optimization, 13, 455–492, 1998.
Lin, M. and Porté-Agel, F.: Power Maximization and Fatigue-Load
Mitigation in a Wind-turbine Array by Active Yaw Control: an
    LES Study, in: Journal of Physics: Conference Series, vol. 1618,
    p. 042036, IOP Publishing, 2020.
López, B., Guggeri, A., Draper, M., and Campagnolo, F.: Wake
    steering strategies for combined power increase and fatigue dam-
age mitigation: an LES study, in: Journal of Physics: Conference
    Series, vol. 1618, p. 022067, IOP Publishing, 2020.
Martínez-Tossas, L. A. and Branlard, E.: The curled wake model:
    equivalence of shed vorticity models, in: Journal of Physics:
    Conference Series, vol. 1452, p. 012069, IOP Publishing, 2020.
Martínez-Tossas, L. A., Annoni, J., Fleming, P. A., and Churchfield,
    M. J.: The aerodynamics of the curled wake: a simplified model
    in view of flow control, Wind Energy Science, 4, 127–138, 2019.
National Renewable Energy Laboratory: WindSE, https://github.
    com/NREL/WindSE, 2021.
Pedregosa, F., Varoquaux, G., Gramfort, A., Michel, V., Thirion,
    B., Grisel, O., Blondel, M., Prettenhofer, P., Weiss, R., Dubourg,
    V., Vanderplas, J., Passos, A., Cournapeau, D., Brucher, M., Per-
    rot, M., and Duchesnay, E.: Scikit-learn: Machine Learning in
    Python, Journal of Machine Learning Research, 12, 2825–2830,
2011.

Picheny, V., Ginsbourger, D., Richet, Y., and Caplin, G.: Quantile-
    based optimization of noisy computer experiments with tunable
    precision, Technometrics, 55, 2–13, 2013.
Qin, C., Klabjan, D., and Russo, D.: Improving the expected im-
    provement algorithm, in: Advances in Neural Information Pro-   30
    cessing Systems, pp. 5381–5391, 2017.
Quick, J., Annoni, J., King, R., Dykes, K., Fleming, P., and Ning,
    A.: Optimization under uncertainty for wake steering strategies,
    in: Journal of physics: Conference series, vol. 854, p. 012036,
    IOP Publishing, 2017.                                          35
Quick, J., King, J., King, R. N., Hamlington, P. E., and Dykes, K.:
    Wake steering optimization under uncertainty, Wind Energy Sci-
    ence, 5, 413–426, 2020.
Rajnarayan, D., Haas, A., and Kroo, I.: A multifidelity gradient-free
    optimization method and application to aerodynamic design, in:  40
    12th AIAA/ISSMO multidisciplinary analysis and optimization
    conference, p. 6020, 2008.
Rasmussen, C. E. and Williams, C. K. I.: Gaussian processes for
    machine learning, Adaptive computation and machine learning,
    MIT Press, Cambridge, Mass, 2006.                              45
Rinker, J. M., Soto Sagredo, E., and Bergami, L.: The Importance
    of Wake Meandering on Wind Turbine Fatigue Loads in Wake,
    Energies, 14, 7313, 2021.
Shahriari, B., Swersky, K., Wang, Z., Adams, R. P., and De Fre-
    itas, N.: Taking the human out of the loop: A review of Bayesian  50
    optimization, Proceedings of the IEEE, 104, 148–175, 2015.

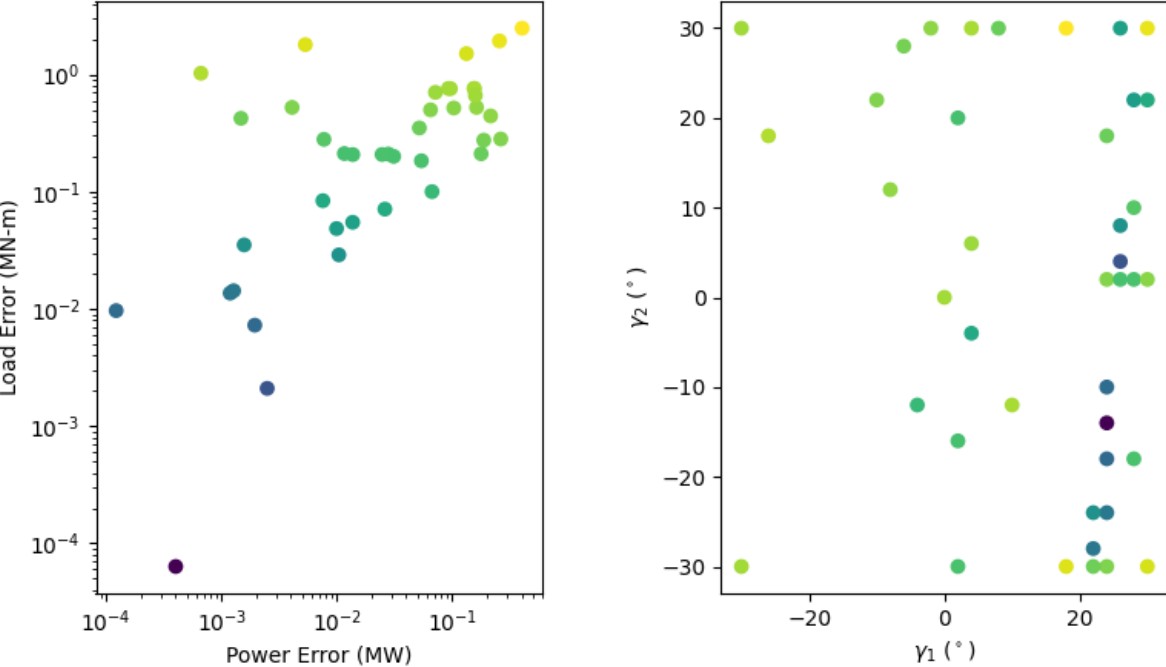

**Figure 10.** Results of the multifidelity leave-one-out analysis. The left panel shows the leave-one-out prediction errors associated with power and loading, and the points are colored by the sum of both errors. The same points are plotted in the right panel, showing their associated $\gamma_1$ and $\gamma_2$ values.

Van Dijk, M. T., van Wingerden, J.-W., Ashuri, T., and Li, Y.: Wind farm multi-objective wake redirection for optimizing power production and loads, Energy, 121, 561–569, 2017.

Virtanen, P., Gommers, R., Oliphant, T. E., Haberland, M., Reddy, T., Cournapeau, D., Burovski, E., Peterson, P., Weckesser, W., Bright, J., van der Walt, S. J., Brett, M., Wilson, J., Millman, K. J., Mayorov, N., Nelson, A. R. J., Jones, E., Kern, R., Larson, E., Carey, C. J., Polat, İ., Feng, Y., Moore, E. W., VanderPlas, J., Laxalde, D., Perktold, J., Cimrman, R., Henriksen, I., Quintero, E. A., Harris, C. R., Archibald, A. M., Ribeiro, A. H., Pedregosa, F., van Mulbregt, P., and SciPy 1.0 Contributors: SciPy 1.0: Fundamental Algorithms for Scientific Computing in Python, Nature Methods, 17, 261–272, https://doi.org/10.1038/s41592-019-0686-2, 2020.

Voorneveld, M.: Characterization of pareto dominance, Operations Research Letters, 31, 7–11, 2003.

Wang, C., Campagnolo, F., and Bottasso, C.: Does the use of load-reducing IPC on a wake-steering turbine affect wake behavior?, in: Journal of Physics: Conference Series, vol. 1618, p. 022035, IOP Publishing, 2020.

Yang, K., Palar, P. S., Emmerich, M., Shimoyama, K., and Bäck, T.: A multi-point mechanism of expected hypervolume improvement for parallel multi-objective bayesian global optimization, in: Proceedings of the Genetic and Evolutionary Computation Conference, pp. 656–663, 2019.

Yin, X., Zhang, W., Jiang, Z., and Pan, L.: Data-driven multi-objective predictive control of offshore wind farm based on evolutionary optimization, Renewable Energy, 2020.

Zalkind, D. S. and Pao, L. Y.: The fatigue loading effects of yaw control for wind plants, in: 2016 American Control Conference (ACC), pp. 537–542, IEEE, 2016.

Zhan, D. and Xing, H.: Expected improvement for expensive optimization: a review, Journal of Global Optimization, 78, 507–544, 2020.