# Peer review of "Multifidelity Multiobjective Optimization for Wake Steering Strategies"

_Wind Energy Science, 2021_

## Referee Comment (RC2)

Review of manuscript: WES-2021-152
Title: Multifidelity multiobjective optimization for wake steering strategies
Authors: Quick, King, Barter & Hamlington

**Overall comments:**

The submitted manuscript proposes a methodology for multifidelity (using multiple models with different cost and accuracy) and multiobjective optimization (with multiple goals, such as high power and low fatigue) for wake steering. The paper is timely, interesting, and satisfies a need for research methods which include consideration of loads in wake steering optimization.

I have several comments and questions listed below that I would like the authors to consider in their revision.

**General comments and questions:**

1. Even a slight change in the downwind turbine location is likely to substantially change the Pareto set results. I am especially looking at the results in Section 4.2 Flow Physics Insights. It seems from Figure 6 that negative yaw misalignment underperforms positive yaw only because of a very slight overlap between the curled wake shape and the lower half of the downwind rotor. I expect small changes in the ABL shear, stability, etc. would also change the results. It would be helpful to more clearly highlight throughout the manuscript that your results are specific to ABL properties and the turbine layout considered in your test case.
2. How does this methodology scale to higher dimensional input spaces (i.e. more than two values of control inputs)? A discussion of the scaling and potential challenges that it brings would be useful to understand how this concept may perform in more realistic scenarios. Especially since this Pareto set would be unique to the wind farm layout, ABL conditions, etc (**Point #1**). So I expect that the Pareto set would need to be uniquely computed over these independent variable input combinations (curse of dimensionality)?
3. I am wondering about two forms of uncertainty not discussed in the manuscript:
    a. Sampling uncertainty - all results are taken from CFD with finite-time averages. Does this impact your results? Relatedly, are all CFD cases started from identical initial conditions? Are your Pareto sets robust to sampling uncertainty?
    b. Meta-uncertainty - How does the meta-uncertainty over different random seeds for your initial sampling points and your initial conditions affect the output Pareto set?
4. The refined sampling points shown in Figure 5 are helpful, but further validation of the proposed methodology's ability to capture the Pareto front would be useful. Can the authors refine their grid search over the $\gamma_1$ and $\gamma_2$ space?

**Point comments:**

1. Line 5: What is meant by "unsteady LES." Is there a time-dependent boundary condition or just turbulent variations about a mean state?

2. Line 19: "A counter-rotating pair of vortices is generated by the rotating blades"
   The counter-rotating pair is also shed by non-rotating turbine models [1,2], so perhaps it is unclear to say that the counter-rotating pair of vortices is generated by 'the rotating blades', but rather 'the yawed rotor'. The rotating blades do also affect the dynamics of the counter-rotating vortex pair [1].

3. Line 28: "Damiani et al. (2018) performed a detailed analysis of a single wind turbine, noting that negative yaw offsets tended to increase fatigue loading more than positive yaw offsets."
   The primary conclusions of Damiani et al. (2018) are that the loading depends on the site conditions (e.g. shear) and turbine model. From the referenced paper conclusions: "On average, the blade-root bending moment DEL decreased for positive yaw offsets and increased for negative offsets. Fairly large variations can be attributed to different turbulence seeds and data records, making generalization more difficult." It is worth providing that context in your statement, since the result of negative yaw leading to more fatigue than positive yaw will not always hold.

4. Line 36: "While engineering wake models are remarkably accurate in power prediction [...]"
   I am not sure the subjective descriptor "remarkably accurate" is useful or true. What level of accuracy is remarkable? Wake models exhibit predictive error in many applications.

5. Line 63: Here you state that the objective is always minimization in this section (Section 2) but subsections 2.1 and 2.2 use maximization objectives.

6. Equation 1: What is the dimensionality of $\gamma$? Is it of size = number of turbines?

7. Line 66: Here $f_i$ is not defined, is that intentional? I was not sure if $g_i(\gamma)$ is emulating $f_i$ directly or is related to $f_i$ somehow through an objective function?

8. Line 73: Related to the note above, on this line, $f_i(\gamma)$ is referred to as an "output" instead of an "objective function." This confusion comes up a few times later in the paper so it is worth clarifying explicitly here.

9. Equation 4: It would be useful to add a validation of the GP model used.

10. Equation 9: Define Pareto dominance

11. Equation 12: $r_1$ and $r_2$ are not defined, I assume they are the coordinates of the reference point in two dimensional space?

12. Section 2.2: The difference between $N$ and $l$ isn't clear in this section

13. Line 155: "No matter which fidelity is to be sampled next, the ultimate goal is to minimize the highest-fidelity function, [...]"
    This is confusing. Here you mean the goal is to minimize the highest fidelity function meaning $f$? But to do that you maximize the objective function J ($f$ also being a different objective function).

14. Equation 18: So there is a different objective J for each model fidelity? That is not clear in this section if so. Also, it is a little confusing to have both J and f as objective functions.

15. Line 180: Is the turbine nacelle or tower included? This has been noted to affect CVP dynamics [3] so what the authors are using should be stated.

16. Figure 1:
    a. The comparative step between the high and low fidelity objectives in the workflow is not explained in the text.
    b. LF and HR (presumably high and low fidelity) not defined.

17. Figure 2: The 'low-fidelity model' looks like it has unphysical grid-to-grid oscillations in the output. How do these unphysical CFD errors impact your results?
18. Line 226: I am puzzled by the authors' choice of normalization to have the objectives be in the same order of magnitude. The choice seems *ad hoc*. Why not use a more precise transformation to ensure they are more directly comparable (e.g. standardization transform). A multiobjective objective function composed of two different units (MW and Nm) seems strange.
19. Line 241: Why will random sampling 'drastically affect the optimization.' What do you mean by 'drastically'? The authors could (perhaps should) account for meta-uncertainty by testing the results over several initialization realizations.
20. Line 243: Missing degree symbol
21. Section 3.3.2 should be mentioned earlier, perhaps in an outline introduction to Section 3. It was confusing as written. Several questions came to mind:
    a. How did the authors specify that 0.89 correlation is sufficiently high while 0.74 (correlation between HF DEL and LF DEL) is not?
    b. Does this correlation depend on the yaw misalignment? In the introduction, the authors stated that the bending moments depend on yaw.
    c. I anticipate that this will depend on the inflow conditions as well, so I am wondering how this method could be used in practice.
22. Section 3.3.2: I am wondering what these results suggest about the approach of 'low-fidelity loads modeling.' It would be helpful to more clearly discuss why the low-fidelity model fails to capture the fatigue. Is the turbulence in the low-fidelity model insufficiently resolved such that it misses the effect of turbulence on the loading?
23. Equation 28: Is the DEL function missing here? In Equation 27, L = DEL(M), not just M.
24. Figure 7:
    a. This figure is very small, please increase the size
    b. I found it to be confusing that the wake deficit increase from x/D=6 to x/D=8, but that is because the downwind turbine is at x/D=7. That should be made more clear in the figure. I am not sure what I am supposed to learn from the x/D=8 contours.
25. Figure 8: Likewise, this figure is small and has many lines. Hard to see.
26. Line 334: "A positive front turbine yaw offset is more effective at reducing loading and increasing power than a negative yaw offset because the counter-rotating vortices produce a greater velocity deficit in the downstream wake."
    I believe this sentence needs to be re-phrased. The authors meant to say that positive yaw leads to less velocity deficit in the wake region (at least the wake region where the downwind turbine is located).

**References**

[1] Howland, Michael F., Juliaan Bossuyt, Luis A. Martínez-Tossas, Johan Meyers, and Charles Meneveau. "Wake structure in actuator disk models of wind turbines in yaw under uniform inflow conditions." *Journal of Renewable and Sustainable Energy* 8, no. 4 (2016): 043301.

[2] Shapiro, Carl R., Dennice F. Gayme, and Charles Meneveau. "Modelling yawed wind turbine wakes: a lifting line approach." *Journal of Fluid Mechanics* 841 (2018).

[3] Zong, Haohua, and Fernando Porté-Agel. "A point vortex transportation model for yawed wind turbine wakes." *Journal of Fluid Mechanics* 890 (2020).

---

## Author Comment (AC1)

**Response to Referee 1**

We greatly appreciate the time taken by the referee to read our manuscript. We have taken into consideration and addressed all comments, questions, and suggestions from the reviewer, and we feel that the revised manuscript is now substantially stronger as a result. Changes made to the text at the request of the reviewer have been highlighted in red in the revised manuscript. In the following, reviewer comments are repeated in italics and our responses are provided in the bulleted sections of text.

*Comments*

1) *In section 3.1, the authors present that 7D rotor diameters were chosen as the spacing between the two turbines, but no justification or references were provided as to why this distance was chosen. Additionally, no reasoning is provided for the chosen wind speed of 7.5 m/s, and the details of the inflow turbulent intensity at hub height are missing. As all of these parameters (turbine spacing, inflow speed, turbulence intensity) would have a significant impact on wake recovery and hence resulting fatigue and power production of the downstream turbines, further clarification on the impact of these parameters on the methodology and results would be interesting to see.*

- We thank the reviewer for pointing out this opportunity for clarification. In revised paper we have clarified that the presented results are specific to the specified atmospheric conditions on P17L368 and P17L379. Further, on P8L189-192, we explain that we wanted to have optimal solutions that were inside the boundaries of allowable yaw offsets. When turbines are spaced tightly, we found that the optimal power was commonly associated with the largest allowable yaw offset of the front turbine, which was a less challenging optimization case. Ultimately, the primary novelty of this paper is the presentation of the applied framework for wake steering. This framework can be applied with different turbine spacings and atmospheric conditions in the future, as we now note on P19L402-403 in the conclusions.

2) *The numerical modeling section could also benefit with the inclusion of performance curves, such as power/rotational speed/pitch against wind speed and yaw angles. By comparing such curves against reference values from the turbine report, it can be confirmed that the turbine and implemented controller in the numerical set-up are operating correctly.*

- On P8L196-197 we now clarify that our analysis assumed constant rotational speed and pitch angles and that we did not have an integrated controller.

3) *The moments in the paper are evaluated by determining the aerodynamic forces along the actuator line elements according to the equation 25. The authors however do not go into further detail about the blade structure and whether the blade material properties and flexibility are accounted for in their simulations. Blade deformation and structural damping could significantly affect the amplitude of stress reversals and hence the resulting fatigue damage. Furthermore, no information is provided as to why only the blade flapwise bending moments are considered in this study, and the edgewise moments and tower loads are not considered.*

- We agree with the reviewer that adding these details will increase the clarity of the paper. We now clarify that the turbine blades are rigid and without a controller on P8L196-197. Because this study is a demonstration of a method, we simply chose the flapwise bending moment for the purpose of providing an illustrative example. We have correspondingly added a note on P9L234-235 that there are several methods available to quantify loading, although we just consider the flapwise bending moment here.

4) *Since both the high-fidelity and low-fidelity simulations are run for the short time durations of 1,200s and 400s, the measure of accuracy of the computed time averaged power production and DEL could suffer from the small sample sizes. Figures 4 and 5 show the output power and loads for all the simulations, however the range of uncertainty of these values is not addressed. The results could benefit from a supplementary figure showing the uncertainty on the computed power and loads, using a statistical tool such as bootstrapping. Additionally, since the flow-through time is reported to be 301 seconds for the turbine set-up, is the duration of 400 seconds of the low-fidelity model sufficient considering initial transients?*

- We agree with the reviewer that there is likely some uncertainty resulting from the finite-time simulations. Regarding the low-fidelity time duration, on P9L225-226, we have added that this cut-in time and the total low-fidelity model time were selected to avoid the effects of the initial transient period while keeping the time required of the low-fidelity simulation low. We have also added text on P9L221-223 explaining that we validated the time intervals used by comparing analysis results after 600-900 s to results after 900-1,200 s, finding a 2.6% relative difference between the computed powers and 4.2% relative difference between the computed DELs.

5) *While formulating the loading objective in line 225, page 9, it is not clear why a factor of '10' is subtracted from the loads.*

- We agree that this could have been clearer. We now clarify on P10L262-263 that this *ad hoc* approach was chosen to ensure that both power and loading were always negative.

6) *Table two summarizes the total power gain for different yaw angles, however it could be interesting to see an analysis on the power production by the individual turbines as well, as shown for loads in Figure 8.*

- We are grateful to the reviewer for providing this feedback and we have now adjusted Table 2 to reflect the front and back turbine power productions.

---

## Author Comment (AC2)

**Response to Referee 2**

We greatly appreciate the time taken by the referee to read our manuscript. We have taken into consideration and addressed all comments, questions, and suggestions from the reviewer, and we feel that the revised manuscript is now substantially stronger as a result. Changes made to the text at the request of the reviewer have been highlighted in red in the revised manuscript. In the following, reviewer comments are repeated in italics and our responses are provided in the bulleted sections of text.

*Major Comments*

1) *Even a slight change in the downwind turbine location is likely to substantially change the Pareto set results. I am especially looking at the results in Section 4.2 Flow Physics Insights. It seems from Figure 6 that negative yaw misalignment underperforms positive yaw only because of a very slight overlap between the curled wake shape and the lower half of the downwind rotor. I expect small changes in the ABL shear, stability, etc. would also change the results. It would be helpful to more clearly highlight throughout the manuscript that your results are specific to ABL properties and the turbine layout considered in your test case*

- We thank the reviewer for this comment and agree that this caveat could have been clearer in the text. We now clarify that the presented results are specific to the specified atmospheric conditions on P17L368 and P17L379. We have also added text on P18L393 to highlight that uncertainty in atmospheric conditions or yaw positions can substantially impact the results of this analysis.

2) *How does this methodology scale to higher dimensional input spaces (i.e. more than two values of control inputs)? – A discussion of the scaling and potential challenges that it brings would be useful to understand how this concept may perform in more realistic scenarios. Especially since this Pareto set would be unique to the wind farm layout, ABL conditions, etc (Point 1). So I expect that the Pareto set would need to be uniquely computed over these independent variable input combinations (curse of dimensionality)?*

- This is an important point and we have modified P5L136 to note that the cost of optimization of the EHVI generally grows exponentially with the number of inputs. We also now note on P5L137-139 that performing the EHVI optimization using a grid search would become computationally prohibitive for higher dimensional design inputs.

3) *I am wondering about two forms of uncertainty not discussed in the manuscript:*
a. *Sampling uncertainty - all results are taken from CFD with finite-time averages. Does this impact your results? Relatedly, are all CFD cases started from identical initial conditions? Are your Pareto sets robust to sampling uncertainty?*
b. *Meta-uncertainty - How does the meta-uncertainty over different random seeds for your initial sampling points and your initial conditions affect the output Pareto set?*

- We agree that uncertainty is an important consideration and we have added text in the conclusions on P18L393 noting that uncertainty will alter the shape of the Pareto set. To address the sampling uncertainty, we have added text on P9L221-223 explaining that we validated the time intervals used in the analysis by comparing results after 600-900 s and results after 900-1,200 s, finding a 2.6% relative difference between the computed powers and 4.2% relative difference between the computed DELs. We have also added text on P8L199-200 to clarify that all simulations were started with the same initial conditions. With respect to

the meta-uncertainty, we agree that a random sampling approach with different initial seeds would be useful. As we now note on P12L312-314, we performed several shorter optimizations as part of the development process using different random initial seeds to confirm that the multifidelity approach consistently outperformed the single fidelity approach.

4) *The refined sampling points shown in Figure 5 are helpful, but further validation of the proposed methodology's ability to capture the Pareto front would be useful. Can the authors refine their grid search over the $\gamma_1$ and $\gamma_2$ space?*

- We agree that a more refined grid search could be used to further refine the Pareto front. However, our use of the grid search in the present study was intended primarily as a demonstration that, after performing the multifidelity multiobjective optimization to determine several points in the Pareto front, the front can be further refined using a targeted grid search. Given this primary objective, which we now state more clearly in the revised paper on P13L321-323, we hope the reviewer agrees that the presented refinement points are sufficient to confirm the validity of the proposed method.

**Detailed Comments**

1) *Line 5: What is meant by "unsteady LES." Is there a time-dependent boundary condition or just turbulent variations about a mean state?*

- We agree that this was a confusing choice of words and we removed the term "unsteady." The boundary conditions are not time-dependant.

2) *Line 19: "A counter-rotating pair of vortices is generated by the rotating blades". The counter-rotating pair is also shed by non-rotating turbine models [1,2], so perhaps it is unclear to say that the counter-rotating pair of vortices is generated by 'the rotating blades', but rather 'the yawed rotor'. The rotating blades do also affect the dynamics of the counter-rotating vortex pair [1].*

- This point was indeed potentially confusing and we have changed the language in the introduction on P1L19-21 to read: "A counter-rotating pair of vortices is generated by the lateral thrust of the wind turbine rotor, which is determined by the yaw offset direction."

3) *Line 28: "Damiani et al. (2018) performed a detailed analysis of a single wind turbine, noting that negative yaw offsets tended to increase fatigue loading more than positive yaw offsets." The primary conclusions of Damiani et al. (2018) are that the loading depends on the site conditions (e.g. shear) and turbine model. From the referenced paper conclusions: "On average, the blade-root bending moment DEL decreased for positive yaw offsets and increased for negative offsets. Fairly large variations can be attributed to different turbulence seeds and data records, making generalization more difficult." It is worth providing that context in your statement, since the result of negative yaw leading to more fatigue than positive yaw will not always hold.*

- This is a useful caveat to include and we added text on P2L28-29 stating that these conclusions were specific to the turbulence seeds used in the study.

4) *Line 36: "While engineering wake models are remarkably accurate in power prediction [...]" I am not sure the subjective descriptor "remarkably accurate" is useful or true. What level of accuracy is remarkable? Wake models exhibit predictive error in many applications.*

- We now clarify on P2L38 that it is Figures 6-11 in the cited work that show remarkable agreement between low- and high-fidelity models when predicting power, with poorer agreement when predicting loading.

5) *Line 63: Here you state that the objective is always minimization in this section (Section 2) but subsections 2.1 and 2.2 use maximization objectives.*

- We agree that it was somewhat jarring to switch from minimization to maximization and we have altered the definition of $J$ in Section 2.2 to call for the minimization of $J$. Since the definition of the $EHVI$ is an intermediate step, we have kept the note that the $EHVI$ should be maximized.

6) *Equation 1: What is the dimensionality of $\boldsymbol{\gamma}$? Is it of size = number of turbines?*

- We have added text on P3L68 to explicitly define the dimension of $\boldsymbol{\gamma}$ to be the number of turbines considered.

7) *Line 66: Here $f_i$ is not defined, is that intentional? I was not sure if is emulating $g_i(\boldsymbol{\gamma})$ directly or is related to $f_i$ somehow through an objective function?*

- We have altered the text to clarify that $f_i$ are the simulated power and loading, making it clearer that $g_i$ are emulating $f_i$.

8) *Line 73: Related to the note above, on this line, $f_i(\boldsymbol{\gamma})$ is referred to as an "output" instead of an "objective function." This confusion comes up a few times later in the paper so it is worth clarifying explicitly here.*

- To avoid confusion, we have altered introduction of Section 2 to avoid referring to $f_i$ as objectives, and instead refer to them as simulated outputs.

9) *Equation 4: It would be useful to add a validation of the GP model used.*

- We have added a note to P3L73-74 that scikit-learn library (Pedregosa et al., 2011) is a well-validated open-source library where the Gaussian Process code has been extensively validated.

10) *Equation 9: Define Pareto dominance*

- We have added a reference to a text on P5L118 that gives a comprehensive definition of Pareto dominance.

11) *Equation 12: $r_1$ and $r_2$ and are not defined, I assume they are the coordinates of the reference point in two dimensional space?*

- We agree that this definition could have been made clearer and we now clarify that $r_1$ and $r_2$ are coordinates of the reference point on P5L130.

12) *Section 2.2: The difference between $N$ and $l$ isn't clear in this section*

- We have changed Equation 18 to use the symbol $i$ instead of $l$, to remain consistent with the notation introduced in the preceding paragraphs of Section 2.2.

13) *Line 155: "No matter which fidelity is to be sampled next, the ultimate goal is to minimize the highest-fidelity function, [...]" This is confusing. Here you mean the goal is to minimize the highest fidelity function meaning $f$? But to do that you maximize the objective function $J$ ($f$ also being a different objective function).*

- We agree that there is some potential for confusion caused by this wording. We have thus changed the sentence to pluralize the references to functions being minimized. We also now refer to $J$ as the multiobjective *acquisition function* instead of as an optimization objective.

14) *Equation 18: So there is a different objective J for each model fidelity? That is not clear in this section if so. Also, it is a little confusing to have both J and f as objective functions.*

- We now clarify on P6L168 that there is a single $J$ objective that is a function of the model fidelity.

15) *Line 180: Is the turbine nacelle or tower included? This has been noted to affect CVP dynamics [3] so what the authors are using should be stated.*

- We clarify on P8L196-197 that only the turbine blades were represented, and the analysis neglects the presence of the tower and nacelle.

16) *Figure 1:*
*a. The comparative step between the high and low fidelity objectives in the workflow is not explained in the text.*
*b. LF and HR (presumably high and low fidelity) not defined.*

- We have added text preceding Figure 1 on P7L175-180 explaining that the comparative step is linked to the multiobjective acquisition function. We also now better connect the notation in the figure to the notation in the paper.

17) *Figure 2: The 'low-fidelity model' looks like it has unphysical grid-to-grid oscillations in the output. How do these unphysical CFD errors impact your results?*

- This oscillations are indeed important and we now clarify on P12L297-300 and P17L372-373 that the oscillations in the model lead to similar oscillations in the moment signals, causing the need for the different low-fidelity loading model.

18) *Line 226: I am puzzled by the authors' choice of normalization to have the objectives be in the same order of magnitude. The choice seems ad hoc. Why not use a more precise transformation to ensure they are more directly comparable (e.g. standardization transform). A multiobjective objective function composed of two different units (MW and Nm) seems strange.*

- We have added text on P10L262-263, explaining that the normalization constants were chosen based on the initial sampling results to get the power and loading to be on a similar scale, and that 10 was subtracted to ensure that the loading objective was negative.

19) *Line 241: Why will random sampling 'drastically affect the optimization.' What do you mean by "drastically"? The authors could (perhaps should) account for meta-uncertainty by testing the results over several initialization realizations.*

- We have removed the word "drastically" from the text to avoid confusion since, as noted in our response to major comment 3, we have added text on P12L312-314 highlighting the invariance of the conclusions in the paper to different initializations.

20) *Line 243: Missing degree symbol*

- We thank the reviewer for pointing this out. The degree symbol has been added.

21) *Section 3.3.2 should be mentioned earlier, perhaps in an outline introduction to Section 3. It was confusing as written. Several questions came to mind: a. How did the authors specify that 0.89 correlation is sufficiently high while 0.74 (correlation between HF DEL and LF DEL) is not? b. Does this correlation depend on the yaw misalignment? In the introduction, the authors stated that the bending moments depend on yaw. c. I anticipate that this will depend on the inflow conditions as well, so I am wondering how this method could be used in practice.*

- We thank the reviewer for raising these issues. We have added a short introduction to Section 3 on P7L182-185, outlining the contents of the section. In response to point a, we did indeed first attempt the optimization without the modified low-fidelity loading function. When we noticed the single-fidelity approach was outperforming the multifidelity approach, we changed the low-fidelity loading model to favor the form with the larger correlation. We added this text on P11L286-288. In response to point b, the presented correlations are with respect to uniform distributions of potential yaw offsets. From this definition, the correlation cannot depend on a specific set of yaw offsets. Finally, in response to point c, we explain in the introduction on P2L40-41 that "In practice, power and loading will likely be optimized in real time using a singular weighted objective."

22) *Section 3.3.2: I am wondering what these results suggest about the approach of 'low-fidelity loads modeling.' It would be helpful to more clearly discuss why the low-fidelity model fails to capture the fatigue. Is the turbulence in the low-fidelity model insufficiently resolved such that it misses the effect of turbulence on the loading?*

- We agree that this is an interesting point that could have been better explained. We now explain on P12L297-300 and P17L372-373 that the lower-order moment functions avoid the influence of the spurious oscillations caused by the low-fidelity loading model.

23) *Equation 28: Is the DEL function missing here? In Equation 27, L = DEL(M), not just M.*

- We have clarified on P12L297-300 that the DEL is purposefully replaced with the lower-order moment functions to avoid the influence of the spurious oscillations caused by the low-fidelity loading model.

24) *Figure 7: a. This figure is very small, please increase the size b. I found it to be confusing that the wake deficit increase from x/D=6 to x/D=8, but that is because the downwind turbine is at x/D=7. That should be made more clear in the figure. I am not sure what I am supposed to learn from the x/D=8 contours.*

- We have enlarged the figure and omitted the X/D=9 plots to save space. We also clarify in the figure caption that the downstream turbine is located at X/D=7.

25) *Figure 8: Likewise, this figure is small and has many lines. Hard to see.*

- We have enlarged the size of Figure 8 to make it easier to read.

26) *Line 334: "A positive front turbine yaw offset is more effective at reducing loading and increasing power than a negative yaw offset because the counter-rotating vortices produce a greater velocity deficit in the downstream wake." I believe this sentence needs to be re-phrased. The authors meant to say that positive yaw leads to less velocity deficit in the wake region (at least the wake region where the downwind turbine is located).*

- We now clarify that the greater velocity deficit is associated with the former strategy (i.e., the negative yaw offset) on P17L381.

**References**

[1] Howland, Michael F., Juliaan Bossuyt, Luis A. Martínez-Tossas, Johan Meyers, and Charles Meneveau. "Wake structure in actuator disk models of wind turbines in yaw under uniform inflow conditions." Journal of Renewable and Sustainable Energy 8, no. 4 (2016): 043301.

[2] Shapiro, Carl R., Dennice F. Gayme, and Charles Meneveau. "Modelling yawed wind turbine wakes: a lifting line approach." Journal of Fluid Mechanics 841 (2018).

[3] Zong, Haohua, and Fernando Porté-Agel. "A point vortex transportation model for yawed wind turbine wakes." Journal of Fluid Mechanics 890 (2020).

We thank the reviewer for these useful comments, and the paper has now been revised in order to address all of the above points. Sincerely, the authors.

---

## Referee Report (RR1)

Review of manuscript: WES-2021-152-R1
Title: Multifidelity multiobjective optimization for wake steering strategies
Authors: Quick, King, Barter & Hamlington

**Overall comments:**

Thank you to the authors for considering the reviewers' comments. The multiobjective optimization methodology proposed by the authors is without doubt a useful, interesting contribution to the wake steering literature. However, I believe the presentation and the interpretation of the results could still be improved. I would like the authors to consider the below comments, with the hope that they can help improve the presentation and the impact of the manuscript's contributions.

**Point comments:**

1. Page 2, Line 26: "Damiani et al. (2018) performed a detailed analysis of a single wind turbine, noting that negative yaw offsets tended to increase fatigue loading more than positive yaw offsets (although it should be noted that these results were specific to the turbulence seeds used in the study)."
   The added parenthetical remark does not provide the necessary caveat. Damiani et al. (2018) specifically cautioned that the influence of the incident conditions "make generalization more difficult." The authors should state that Damiani et al. (2018) cautioned against the general statement that "negative yaw offsets tended to increase fatigue loading more than positive yaw offsets."

2. Page 2, Line 37: I am still concerned with the phrasing "remarkably accurate in power prediction." This seems to highlight that wake modeling for power predictions is a concluded matter. I don't believe this is the case, as wake models can exhibit high predictive error for many utility-scale wind farm applications, e.g. time-varying conditions, the presence of complex terrain, strong stratification, etc., and this statement is counter to the community calls for research [e.g. 1, 2].

3. Page 3, Line 73: "We use the scikit-learn Gaussian Process implementation (Pedregosa et al., 2011), which is a well-validated open-source project."
   It's useful that the authors have added this, but specifically, I would like to see: 1) a comparison of the GP fit to the training data; 2) a GP prediction (out of sample of the training data) compared to the simulated power and loads for an out of sample set of the design variables.

4. Page 5, Line 118: Many readers of *Wind Energy Science* will not be familiar with Pareto dominance, and this article should be self-contained. Please add one or two sentences defining Pareto dominance before referring to the paper or remove its use from the manuscript.

5. Page 9, Line 221: "These time parameters were justified by comparing power and loading computed over time intervals of 600-900 s and 900-1,200 s, resulting in relative differences of

only 2.6% for power and 4.2% for loading when \gamma = (15;0)."
These differences seem nontrivial. Have the authors tested how sensitive the Pareto set is to these finite-time averages? Does the optimal yaw change 0.1 degrees or 10 degrees? Especially given how sensitive the flow physics interpretation is to the particular wind conditions, I am interested in hearing more about this.

6. Eq. (32): Now that you have clarified that the choice of the objective function (especially the -10 in the loads function) was *ad hoc* to have a negative value, I am wondering about the effect of your objective function here. Is the particular quantity of the objective function entirely arbitrary in your framework or will it impact your Pareto front and/or your exploration/exploitation tradeoff? Perhaps the mean of the objective does not matter but it is the sensitivity with respect to the design variables which matters? Simply stated, a reader will want to know: how do I pick an objective function if I want to apply this proposed method? Should it depend on the turbine model, farm geometry, etc.?

7. It is reasonable to ask whether the presented results actually show that the multifidelity approach is better than the single fidelity. For minimizing EHVI, it seems that multifidelity is better (although the noise is larger for multifidelity and if you extrapolate the trend lines it would seem to suggest the blue line will drop below the orange with more evaluations).
For the end objectives of power and loads, the multifidelity is only faster to estimate minimum loads, it is actually slower for estimating maximum power. One also has to recognize that these are the results where the authors have intentionally created an artificial loads model such that the multifidelity is better than the single fidelity (Page 11, Line 286). Does this bias the results?
I would appreciate for the authors to confront this more directly. The paper is clearly a novel, significant contribution already based on the multiobjective optimization approach. But can it be concluded that multifidelity is superior to single fidelity? Perhaps this is a call to action for better loads modeling more than anything.

**References**

[1] Veers, Paul, Katherine Dykes, Eric Lantz, Stephan Barth, Carlo L. Bottasso, Ola Carlson, Andrew Clifton et al. "Grand challenges in the science of wind energy." *Science* 366, no. 6464 (2019): eaau2027.

[2] Meneveau, Charles. "Big wind power: seven questions for turbulence research." *Journal of Turbulence* 20, no. 1 (2019): 2-20.

---

## Author Response (AR2)

**Response to Referee 2**

We greatly appreciate the time taken by the referee to read our manuscript. We have taken into consideration and addressed all comments, questions, and suggestions from the reviewer, and we feel that the revised manuscript is now substantially stronger as a result. Changes made to the text at the request of the reviewer have been highlighted in red in the revised manuscript. In the following, reviewer comments are repeated in italics and our responses are provided in the regular sections of text.

**Point Comments**

**1.** *Page 2, Line 26: "Damiani et al. (2018) performed a detailed analysis of a single wind turbine, noting that negative yaw offsets tended to increase fatigue loading more than positive yaw offsets (although it should be noted that these results were specific to the turbulence seeds used in the study)." The added parenthetical remark does not provide the necessary caveat. Damiani et al. (2018) specifically cautioned that the influence of the incident conditions "make generalization more difficult." The authors should state that Damiani et al. (2018) cautioned against the general statement that "negative yaw offsets tended to increase fatigue loading more than positive yaw offsets."*

We agree that this was too general of a statement. We have revised the manuscript on P2L26-31 to clarify that Damiani et al. (2018) stressed against generalizing the findings of their single turbine study. We have also now added references to studies of wake steering effects on downstream turbines. In particular, we point to Figures 2 and 4 in (López et al., 2020) and Figure 3 in (Zalkind and Pao, 2016).

**2.** *Page 2, Line 37: I am still concerned with the phrasing "remarkably accurate in power prediction." This seems to highlight that wake modeling for power predictions is a concluded matter. I don't believe this is the case, as wake models can exhibit high predictive error for many utility-scale wind farm applications, e.g. time-varying conditions, the presence of complex terrain, strong stratification, etc., and this statement is counter to the community calls for research [e.g. 1, 2].*

We agree that this phrasing was somewhat overzealous. We removed this sentence to avoid causing confusion. We now only claim that engineering wake models have dubious accuracy when predicting fatigue loading on P2L38-40.

**3.** *Page 3, Line 73: "We use the scikit-learn Gaussian Process implementation (Pedregosa et al., 2011), which is a well-validated open-source project." It's useful that the authors have added this, but specifically, I would like to see: 1) a comparison of the GP fit to the training data; 2) a GP prediction (out of sample of the training data) compared to the simulated power and loads for an out of sample set of the design variables.*

These are important points and we have consequently conducted a leave-one-out analysis to assess the accuracy of the single-fidelity (SF) and multifidelity (MF) GP models, excluding one point at a time to compare the prediction of the GP model to the observed value. Excluding some points makes their associated predictions outside of the training range. The results and a corresponding discussion were added to a new appendix at the end of the revised paper, also shown in Figures 1 and 2 at the end of this document. Generally, there is slightly more error incurred by the prediction of the MF GP than the SF GP. We reference the results in the appendix on P14L343-P15L344.

**4.** *Page 5, Line 118: Many readers of Wind Energy Science will not be familiar with Pareto dominance, and this article should be self-contained. Please add one or two sentences defining Pareto dominance before referring to the paper or remove its use from the manuscript.*

We have added a brief description of what Pareto dominance means on P5L119-120.

**5.** *Page 9, Line 221: "These time parameters were justified by comparing power and loading computed over time intervals of 600-900 s and 900-1,200 s, resulting in relative differences of only 2.6% for power and 4.2% for loading when $\gamma = (15; 0)$." These differences seem nontrivial. Have the authors tested how sensitive the Pareto set is to these finite-time averages? Does the optimal yaw change 0.1 degrees or 10 degrees? Especially given how sensitive the flow physics interpretation is to the particular wind conditions, I am interested in hearing more about this.*

We agree that the temporal convergence of the underlying simulation is an important concern. To further test this convergence, we performed longer simulations at each of the sample points during the optimization, and Figure 3 at the end of this document compares the Pareto fronts found using time intervals of 600-1,200 seconds and 1,200 seconds-1,800 seconds. Although the shape of the Pareto front does change slightly, the hypervolume of the Pareto fronts discovered by the single-fidelity and multifidelity optimizations each differed by less than 0.5% between the two analysis periods. We have added text summarizing this further analysis on P9L223, and we have also added to the conclusions on P19L410-412 that care should be taken when applying this method to ensure convergence of the Pareto set with respect to convergence of the underlying simulation.

**6.** *Eq. (32): Now that you have clarified that the choice of the objective function (especially the -10 in the loads function) was ad hoc to have a negative value, I am wondering about the effect of your objective function here. Is the particular quantity of the objective function entirely arbitrary in your framework or will it impact your Pareto front and/or your exploration/exploitation tradeoff? Perhaps the mean of the objective does not matter but it is the sensitivity with respect to the design variables which matters? Simply stated, a reader will want to know: how do I pick an objective function if I want to apply this proposed method? Should it depend on the turbine model, farm geometry, etc.?*

We have clarified on P11L269-271 that, because the EHVI is an area produced by the two objectives, we do not expect different values in this scaling function to affect the results of maximizing the acquisition function, provided that all sampled objective values are always less than the associated reference value.

We note that this scaling must be done in *ad hoc* manner, as there is no "optimal" scaling of the two functions that we are aware of, and having the objectives all have the same goal of minimization simplifies the problem formulation. The scaling should be selected so that the functions are of similar magnitude and will never change sign. Since the expected hypervolume improvement acquisition function is based on the area formed by the two objectives, we do not expect the magnitude of the individual objectives to have a great influence on the optimal choice. For example, we internally confirmed that using a different load scaling (i.e., $L = \hat{L}/10^6 - 15$) results in the same solution of which set of yaw offsets to sample next in a multifidelity case.

**7.** *It is reasonable to ask whether the presented results actually show that the multifidelity approach is better than the single fidelity. For minimizing EHVI, it seems that multifidelity is better (al-*

*though the noise is larger for multifidelity and if you extrapolate the trend lines it would seem to suggest the blue line will drop below the orange with more evaluations). For the end objectives of power and loads, the multifidelity is only faster to estimate minimum loads, it is actually slower for estimating maximum power. One also has to recognize that these are the results where the authors have intentionally created an artificial loads model such that the multifidelity is better than the single fidelity (Page 11, Line 286). Does this bias the results? I would appreciate for the authors to confront this more directly. The paper is clearly a novel, significant contribution already based on the multiobjective optimization approach. But can it be concluded that multifidelity is superior to single fidelity? Perhaps this is a call to action for better loads modeling more than anything.*

We agree that our original draft reads as if the power converged faster in the single-fidelity case, which would be an unintuitive result. We removed text on P12L310-313 that had erroneously claimed the single-fidelity optimization optimal power converged faster than the multifidelity counterpart, when, in fact, the single fidelity approach had only converged to the *approximate* optimal solution faster. Close inspection of Figure 3 in the paper shows that the power for the single-fidelity approach converges to the best solution later than in the multifidelity case.

We also stress that the most important convergence metric – indeed, the ultimate goal of the present approach – is to achieve convergence of the hypervolume spanning multiple objectives. That is, we can only claim true convergence once *all* objectives (i.e., both loading and power) have been optimized. In this sense, the single-fidelity approach is less efficient because it takes longer than the multi-fidelity approach to converge for all objectives.

Nevertheless, as with the Damiani *et al.* (2018) study, we caution against generalizing the present results to all wind farm optimization sutides. We have added text to the conclusions on P18L386-387 confronting the possibility of bias and calling for this approach to be applied to more wind farm layouts to assess if the single-fidelity approach consistently finds the optimal power production faster than the multifidelity approach.

**Appendix**

[Figure]

Figure 1: Results of single-fidelity LOO. The left panel shows the leave-one-out prediction errors associated with power and loading, and the points are colored by the sum of both errors. The same points are plotted in the right panel, showing their associated $\gamma_1$ and $\gamma_2$ values.

[Figure]

Figure 2: Results of multifidelity LOO. The left panel shows the leave-one-out prediction errors associated with power and loading, and the points are colored by the sum of both errors. The same points are plotted in the right panel, showing their associated $\gamma_1$ and $\gamma_2$ values.

[Figure]

Figure 3: Pareto fronts discovered by the single-fidelity and multifidelity optimization algorithms, computed using the original time wind (600-1,200 seconds, triangles) and a new time window (1,200-1,800 seconds, crosses). The plot in the left column results associated with the single fidelity optimization and the plot in the right column shows the results associated with the multifidelity optimization.

---

## Author Response (AR3)

**Response to Referee**

We greatly appreciate the time taken by the referee to read our manuscript. We have taken into consideration and addressed all comments, questions, and suggestions from the reviewer, and we feel that the revised manuscript is now substantially stronger as a result. Changes made to the text at the request of the reviewer have been highlighted in red in the revised manuscript. In the following, reviewer comments are repeated in italics and our responses are provided in the regular sections of text.

*"One final comment is that I personally found the 'leave-one-out' analysis results to be unclear, especially since the error units are dimensional. It is not immediately clear from Figures 9 and 10 whether the GP is indeed reliable (e.g. 0.1 MW error in Region 2 operation when only a single training point is excluded appears to be significant). Perhaps consider reformatting the plot or contextualizing these errors."*

We added contextual information to the appendix. We now note that many of the sampled errors are less than 0.1 MW and 0.1 MN-m, particularly in the region of the discovered Pareto set, which correspond to 3% of the maximum power and 6% of the minimum loading, respectively. We also added a note that the multifidelity approach yielded higher maximum errors and lower minimum errors than the single-fidelity approach.